# Benchmarking Diversity in Text-to-Image Models via Attribute-Conditional Human Evaluation

**Isabela Albuquerque**
Google DeepMind

**Ira Ktena**
Google DeepMind

**Olivia Wiles**
Google DeepMind

**Ivana Kajić**
Google DeepMind

**Amal Rannen-Triki**
Google DeepMind

**Cristina Vasconcelos**
Google DeepMind

**Aida Nematzadeh**
Google DeepMind

## Abstract

Despite advancements in photorealistic image generation, current text-to-image (T2I) models often lack diversity, generating homogeneous outputs. This work introduces a framework to address the need for robust diversity evaluation in T2I models. Our framework systematically assesses diversity by evaluating individual concepts and their relevant factors of variation. Key contributions include: (1) a novel human evaluation template for nuanced diversity assessment; (2) a curated prompt set covering diverse concepts with their identified factors of variation (e.g. prompt: AN IMAGE OF AN APPLE, factor of variation: color); and (3) a methodology for comparing models in terms of human annotations via binomial tests. Furthermore, we rigorously compare various image embeddings for diversity measurement. Our principled approach enables ranking of T2I models by diversity, identifying categories where they particularly struggle. This research offers a robust methodology and insights, paving the way for improvements in T2I model diversity and metric development.

## 1 Measuring diversity in text-to-image models

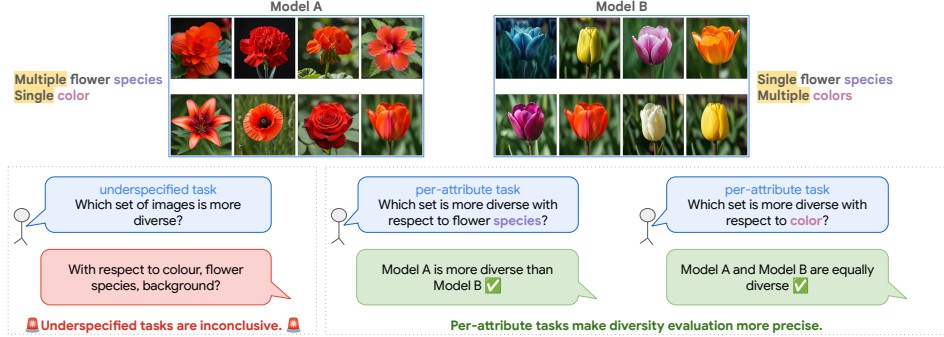

Figure 1: Evaluating diversity requires specifying both the concept being assessed and the factor of variation to reduce ambiguity in the annotation process.

Output diversity is widely considered desirable for text-to-image (T2I) generation models aiming to accurately represent the natural variability of entities in the real world. This is crucial not only technically, for serving as faithful world models, but also for downstream applications like supporting creative processes and ensuring broad conceptual representation across contexts. For example, a diverse model generating "an image of a house" should produce variations in architectural style and background. However, current diversity metrics often conflate it with other properties like fidelity

(e.g., Fréchet Inception Distance (FID) [14]). While progress has been made by developing dedicated metrics (e.g., Vendi Score [10]), the conditions for measuring diversity remain poorly defined and lack standardization, highlighting the need for a principled framework.

In particular, previous work often measures the variability of generated images in scenarios that do not explicitly account for diversity. For instance, images may be generated using a prompt set that neither requires nor controls for output variations [e.g., 31, 1], or models may be compared using a generic human evaluation template that does not specifically probe for diversity [e.g., 3]. This can result in measures of diversity that are ambiguous or inconclusive (see Fig. 1). To address this challenge, we propose a framework to measure diversity without conflating constructs [43, 44, 25, 16, 41]: we operate under the premise that systematically evaluating diversity requires specifying both the concept being assessed and the attribute of interest, as illustrated in Fig.1. We empirically validate this by demonstrating that human accuracy in evaluating diversity is at chance level when the attribute is not defined. Building on this observation, we introduce a novel evaluation framework designed to measure the per-attribute intrinsic diversity of T2I models. This framework includes a synthetically generated prompt set spanning common concepts and their variations, as well as a human evaluation template. The template, informed by empirical findings on a golden set, improves human accuracy by dividing the evaluation into two subtasks: counting and counts comparison.

Considering the high cost of human evaluations for model ranking, developing automated metrics that accurately reflect human judgment is crucial for advancing T2I models. While various diversity metrics have been proposed [10, 16], their alignment with human perceptions of diversity often remains unevaluated. To address this, we use our proposed human evaluation template and prompt set to examine the reliability of autoevaluation metrics. Specifically, we investigate the Vendi Score [10], a widely adopted diversity metric [19, 12] whose correlation with human-perceived diversity has not yet been thoroughly established. Our analysis reveals that the Vendi Score, when optimized for the appropriate representation space, can achieve approximately 65% accuracy in capturing human diversity judgments. We also find that the accuracy improves to 80% when the model pairs are more different, highlighting the need for more discriminant representations. Furthermore, we apply our framework to compare five recent generative models: Imagen 3 [2], Imagen 2.5 [39], Muse 2.2 [5], DALLE3 [3], and Flux 1.1 [20]. This comparison identifies Imagen 3 and Flux 1.1 as the top-performing models regarding attribute diversity. We believe our framework provides a robust foundation for future work in developing more human-aligned evaluation metrics and improving T2I model diversity. This research makes three key contributions:

- It formalizes the problem of quantifying diversity in T2I models and proposes a practical evaluation approach using pre-defined factors of variation.
- It introduces an evaluation framework consisting of a detailed prompt set (covering 86 concept-factor variation pairs) and a validated human evaluation template.
- It applies this framework to evaluate prominent T2I models and automatic evaluation metrics.

## 2    The three ingredients for diversity evaluation

To evaluate diversity, our framework is based on three components: a definition of what specific diversity is being measured, a prompt set to elicit relevant outputs, and a human evaluation template for reliably comparing models. These are described below.

### 2.1    A clearly specified problem: Diversity per attribute

**Prelude: formalizing diversity.** Consider a set of images $X = \{x_1, x_2, \ldots, x_n\}$, where each image $x_i$ belongs to a space $\mathcal{X} \subseteq \mathbb{R}^D$. We posit that the visual appearance of each image $x_i$ is primarily determined by a set of $K$ underlying independent generative factors $f_i = \{f_i^1, \ldots, f_i^K\}$. A potential generative model could be formulated as:

$$p(x_i) = \prod_{k=1}^{K} p(x_i|f_i^k)p(f_i^k). \tag{1}$$

We focus on scenarios where images represent scenes containing instances from well-defined concepts (e.g., bottle, forest). Given a concept, we can often map these abstract generative factors to concrete, observable attributes. For instance, an image $x_i$ depicting a bottle can be described by attributes such as: $f^{\text{material}} \in \{\text{glass, plastic, metal}\}$, $f^{\text{shape}} \in \{\text{cylindrical, square}\}$, and $f^{\text{state}} \in \{\text{open, closed}\}$.

Let $C = \{c^1, \ldots, c^J\}$ be the set of concepts, $A^j = \{a^{j,1}, \ldots, a^{j,K}\}$ the relevant attributes for a given concept $c^j$, and $V^{j,k}$ the finite set of possible values for attribute $a^{j,k}$. Each image $x_i$ depicting a concept is associated with a specific value $v_i^{j,k} \in V^{j,k}$ for each attribute $a^{j,k}$. We define a sample of images $X^j$ (for the same concept $c^j$) as *perfectly diverse* if it comprehensively covers all attribute variations. More precisely, for every attribute $a^{j,k} \in A^j$ and every possible value $v \in V^{j,k}$ there must exist at least one image $x_i^j \in X^j$ such that the attribute $a^{j,k}$ for image $x_i^j$ takes the value $v$.

**A tractable notion of diversity.** Measuring diversity across the complete set of generative factors underlying natural data is significantly challenging. Firstly, the sheer number of potential factors ($K$) is often immense. Secondly, as highlighted by Tsirigotis et al. [38], the combination of their possible values grows exponentially, leading to a 'curse of generative dimensionality' where no realistic finite sample can cover all possible combinations. Thirdly, many factors may inherently possess continuous value ranges, making exhaustive coverage impossible even for a single factor.

Given these challenges, and since achieving the *perfect diversity* (as defined earlier) is intractable with a finite sample, we instead propose to measure *tractable diversity*. This approach focuses on a carefully selected subset of the most salient and practically relevant generative factors ($K'$) for a specific concept. Identifying which factors are practically relevant is non-trivial and must be tailored for a given use case. In this work, to identify these factors, we focus on commonly observed concepts reflective of T2I model training data. To effectively sample from the distribution of generative factors within these concepts, we leverage the knowledge encoded by Large Language Models (LLMs) [30]. Specifically, we prompt an LLM (Gemini 1.5 M [37]) to identify relevant aspects of variation for evaluating the diversity of a given concept. The full system instruction is given in the Appendix.

## 2.2 A systematically generated prompt set

Our goal is to rigorously evaluate generative models and diversity metrics, specifically focusing on their ability to represent variation within distinct attributes of concepts. To effectively rank these models and metrics, our evaluation framework must accommodate both precisely controlled scenarios and complex, real-world use cases. We deliberately select concepts that are ubiquitous in everyday life and common image datasets, such as ImageNet [8] (e.g., 'fruit', 'car', 'snake'), thereby anchoring our evaluation in practical utility. However, simple concepts alone are insufficient. They must also possess inherent complexity and variability, presenting a genuine challenge to the models and metrics. The chosen concepts and their attributes need to be sufficiently nuanced to allow our evaluation methodology to clearly reveal performance differences and track improvements over time or across different systems.

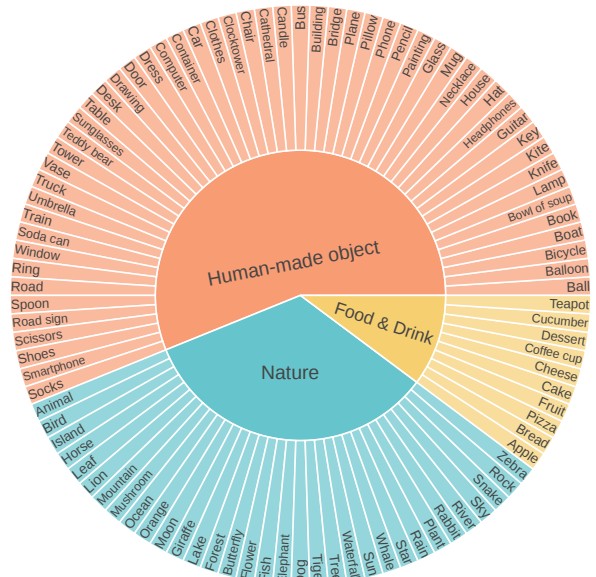

Figure 2: Each slice represents a concept, grouped and color-coded by its overall category.

To structure this process, we classify concepts into three widely applicable categories: *Food and Drink* (items like *coffee cup*, *cake*), *Nature* (elements like *river*, *butterfly*), and *Human-made Objects* (artefacts like *bridge*, *laptop*). We leverage the generative capabilities of Large Language Models (LLMs) to systematically produce a wide range of concepts within these categories. The instruction to generate "ImageNet-like" concepts guides the LLM towards producing concrete, typically visualizable nouns, similar in scope to those in large-scale image datasets. For each generated concept, we then perform a subsequent step, again using an LLM, to identify a semantically relevant aspect of variation (attribute) that is intrinsic or commonly associated with that concept. This yields concept-attribute pairs $(c^j, a^{j,k})$ such as: (*apple*, *color*), (*car*, *type*),

(*tree*, *species*), (*coffee cup*, *material*), (*chair*, *style*). This two-stage, LLM-driven process allows us to systematically build a prompt set specifically designed to probe and evaluate diversity along meaningful, contextually relevant dimensions for a broad range of common concepts. Finally, the authors manually verified all concept-attribute pairs and removed 5 where the attribute was potentially difficult / ambiguous to categorize (e.g. (*food*, *cuisine*)).

## 2.3 A validated, bespoke human evaluation template

Prior work has shown that developing an appropriate human evaluation template is an essential component in the process of measuring a desired capability of a generative model [42, 7]. To that end, we develop a human evaluation template that: (a) allows annotators to understand the task well, (b) captures their judgment faithfully, and (c) yields meaningful ground truth annotations for per-attribute diversity, subsequently used to validate automated evaluation metrics. The annotators are provided with 4 options for the side-by-side comparison: (i) Left more diverse, (ii) Right more diverse, (iii) Equally diverse, (iv) Unable to answer.

**A template to measure per-attribute diversity.** Our template for measuring per-attribute diversity employs a comparative, side-by-side approach due to the difficulty of evaluating diversity within a single set. Many existing diversity metrics also require a reference set. We considered the following design choices for our human evaluation template to ensure meaningful assessment (1) *Set size*: Balancing the perception of diversity with minimizing annotation fatigue and enabling robust computation for metrics requiring larger sets (e.g., Vendi score). (2) *Attribute specification*: Explicitly stating the attribute for evaluation versus allowing open-ended diversity assessment. (3) *Anchoring task*: Incorporating an intermediate task to guide annotators to focus on the intended attribute.

**Validating the template with a golden set.** To evaluate the quality of the evaluation template, we curate a golden set of 10 `<concept, aspect>` pairs, where `concept` corresponds to a concept that should be considered common across images in a set and `aspect` describes the associated aspect of variation that we want to measure diversity against. We validate the evaluation template by comparing cases where (i) the concept *remains constant* across images in the set while the aspect *varies* (ii) the concept *varies* across images while the aspect *remains the same*, and (iii) *both* the concept and the aspect vary across images within the set. We expect images in set (i) to be considered more diverse than images in set (ii), and similarly images in set (iii) to be considered more diverse than images in set (ii). Finally, we expect that images in sets (ii) and (iii) are considered equally diverse as we want to focus on the `aspect` as axis of variation.

In Fig. 3, we present the annotation accuracy of human experts using our template under various conditions, treating our definitions (in the previous paragraph) as ground truth. The different templates are shown in Fig. 9. The accuracy for the `w/o aspect` task is 30.0% for comparisons of sets of size 4 and 26.7% for sets of size 8. In contrast, the template that includes the `aspect` shows a significant increase in accuracy (82.5% for set size 4 and 53.3% for set size 8), indicating that explicitly mentioning the desired aspect of variation improves annotation accuracy. This improvement likely stems from preventing annotators from unintentionally conflating the `concept` and the `aspect` when not guided to focus

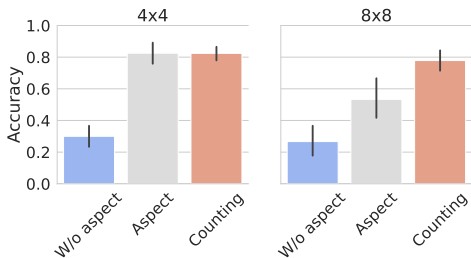

Figure 3: Match with the golden set depending on different set sizes.

on a specific axis. Furthermore, we observe that adding the `count` anchoring question enhances accuracy, especially for the set size of 8, reaching 77.9%.

For the `count` task, we found a strong ($\rho = 0.88$) and statistically significant ($p < .001$) correlation between the annotators' final diversity comparison and the comparison inferred from their individual subset counts (where a higher count on one side implies a `more diverse` final response for that side, and equal counts imply `equal` diversity). This confirms that the `anchoring` count question effectively guides annotators. To further validate our setup, we analyzed instances where annotators' responses deviated from the ground truth in our golden set. We examined the distributions of attribute counts for two image subsets: (1) those labelled "diverse" in the ground truth, where we expected a count mode of "8" and (2) those labelled "non-diverse", where we expected a mode of "1". The

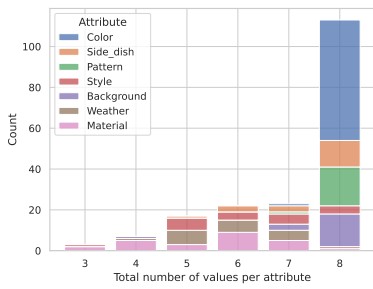

(a) The "diverse" golden set.

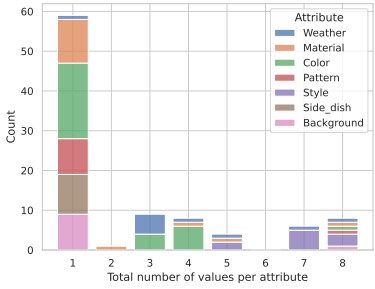

(b) The "non-diverse" golden set.

Figure 4: The distribution of counts for sets of images labelled as "diverse" or "non-diverse" in the golden set for the pilot study.

results of this analysis are presented in Fig. 4. While generally, annotator responses aligned with the golden set labels, we observed a few exceptions. For instance, in one case labelled as a diverse set of chairs, all annotators counted only 3 or 4 distinct chair types, indicating lower diversity than expected. Upon closer inspection, these chairs appeared visually similar despite potentially different underlying material prompts (e.g., metal, iron, aluminum).

# 3 Our framework in practice

We demonstrate our framework's practical application by: (i) collecting comprehensive human annotations with our template to compare models, (ii) using these annotations as ground truth to evaluate diversity metrics, and (iii) comparing model rankings from human versus automatic evaluations to highlight the gap between human-perceived diversity and current metric capabilities.

## 3.1 Ranking models via human evaluation

With the proposed prompt set from Sec. 2.2 and the human evaluation template introduced in Sec. 2.3, we evaluate the attribute-based diversity of five generative models, namely: Muse 2.2 [5], Imagen 2.5 [39], Imagen 3 [2], DALLE3 [3], and Flux 1.1 [20]. For each model, we generate 20 distinct samples for each prompt, randomly combine them in 10 different sets of 8 images, and run side-by-side evaluations for all 10 combinations of 2 models. For each side-by-side comparison, evaluations from 5 different raters were collected. Raters had access to a slide deck with instructions to perform the task and were compensated for the time invested in the data collection. Details can be found in the Appendix (Sec.B) Before comparing each model pair in terms of diversity, we evaluate the overall annotations quality by computing the inter-annotator agreement via Krippendorff's alpha reliability ($\alpha$) [11]. In Fig. 5a, we observe that for all cases $\alpha > 0.8$, indicating a high-degree of agreement across annotators [24].

**Ratings aggregation.** Given the high levels of inter-annotator agreement for all runs of the human evaluation, we aggregate annotations for each side-by-side comparison across raters by *taking the mode* of the ratings. We then follow this step with a second aggregation, this time at the level of all side-by-side comparisons for each concept. For instance, when comparing a given model pair, there are 10 side-by-side comparisons for the concept *apple* (each side-by-side comparison here corresponds to the evaluation of two sets of 8 images). At the end of this process, for the considered models pair, we obtain a single human evaluation result for each concept in the prompt set.

**Model ranking.** Using the results from the ratings aggregation, we propose to use Binomial tests to verify the following hypothesis: *there is a significant difference between the outcomes of a given pair of models*. To do so, we count the number of categories for which each model was deemed best and perform a two-sided Binomial test under the null-hypothesis that the rate for which each model is the best for a concept is equal to 50% (i.e. both models have equal win rates). Results considering a 95% confidence level for all tests are shown is Fig. 5b. Imagen 3 and Flux 1.1 are significantly better or not worse than all other models. Imagen 2.5 and Muse 2.2 are not significantly better than any contender, showing that our benchmark is able to capture an overall progress in diversity when comparing newer and older models. DALLE3 is significantly better than Imagen 2.5, but does not significantly surpass the performance of the other models considered for comparison.

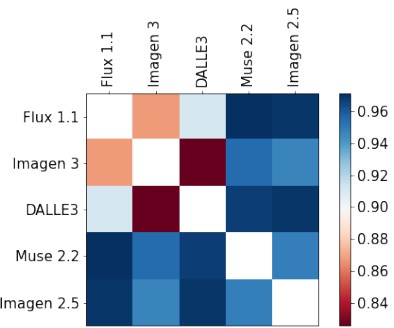 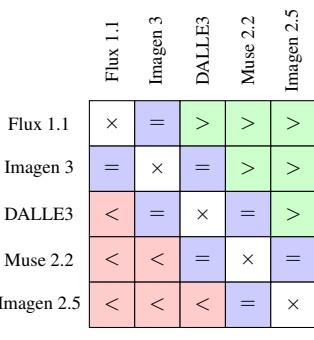

| (a) Krippendorff's $\alpha$-reliability. | (b) Binomial test results at a 95% confidence level. |

Figure 5: **Human evaluation results.** (a) Inter-annotator agreement results in terms of Krippendorff's $\alpha$-reliability. (b) We compare model rankings in terms of significance in the number of wins with two-sided Binomial tests under a 95% confidence level. Each entry in the grid represents a comparison between two models. The sign indicates the model in the row is better ($>$), worse ($<$), or not significantly different ($=$) than the model in the column.

## 3.2 Comparing autoevaluation metrics

While human evaluation is often considered gold standard, it can be impractical to rely solely on human annotation. We then leverage the collected human annotations to perform an extensive study of the role of embeddings for the Vendi Score [1].

**Autoraters based on the Vendi Score.** Given a set of images $X^{j,k} = \{x_i^{j,k}\}$ (corresponding to a given model, concept $c^j$ and attribute $a^{j,k} \in A^j$), we extract embeddings $h_\Xi(x_i^{j,k})$ for each image. $h_\Xi$ is a pretrained feature extractor that can be dependent on a set of conditions $\Xi = \{\xi_l\} \subset (C \times A) \cup \{\xi^0\}$ where $\xi^0$ is a condition unrelated to the considered categories and attributes that can be added to test the impact of conditioning. The different feature extractors and conditions we used are detailed in the following paragraph, but here are a few generic examples to clarify the notation: (i) $h_\Xi$ takes only images as input. In this case, $\Xi = \emptyset$. (ii) $h_\Xi$ is a vision and language model. In this case, embeddings can be conditioned on text data that depends on the concept only (i.e., $\Xi = \{c^j\}$), attribute only (i.e., $\Xi = \{a^{j,k}\}$), or both concept and attribute (i.e., $\Xi = \{c^j, a^{j,k}\}$). To test the impact of conditioning on text, we can instead choose an unrelated prompt (i.e., using $\Xi = \{\xi^0\}$). Finally, we aggregate the embeddings using a diversity metric to obtain a score for the set. As we do not have access to a reliable reference in our setting, we use the Vendi Score [10], a reference-free and widely adopted metric [28, 16, 12, 18]. The Vendi Score is defined as follows:

**Definition 1** (Adapted from [10], Definition 3.1). *Given a concept $c^j$, an attribute $a^{j,k}$ and a set of conditions $\Xi$, let $\{x_1^{j,k}, \ldots, x_n^{j,k}\}$ denote a set of images representing a given concept and attribute. Let $k : X \times X \to \mathbb{R}$ be the cosine similarity between the embeddings of two images, $K^\Xi \in \mathbb{R}^{n \times n}$ be the kernel matrix, with $K_{lm}^\Xi = k^\Xi(x_l^{j,k}, x_m^{j,k})$, and let $\lambda_1^\Xi, \ldots, \lambda_n^\Xi$ be the eigenvalues of $K^\Xi/n$. The Vendi Score for the set $\{x_1^{j,k}, \ldots, x_n^{j,k}\}$ is defined as:*

$$s_\Xi(x_1^{j,k}, \ldots, x_n^{j,k}) = \exp(-\sum_{i=1}^{n} \lambda_i^\Xi \log \lambda_i^\Xi). \tag{2}$$

**Experimental setup.** We compare three different types of embeddings. First, we compare embeddings obtained *using only* the image input. Here we consider two models trained for IMAGENET classification – the IMAGENET INCEPTION model introduced in [36] and an IMAGENET VIT-B/16 model trained on IMAGENET21K as described in [35]. We also consider one self-supervised model, DINOV2 [26]. Second, we consider embeddings conditioned on both the image and textual attribute. We use PALI embeddings [4] at various points after fusing the text and visual input, and CLIP [29] combined text and image embedding. We use these embedding models to obtain an embedding for

---
[1]Results with other autoraters can be found in the Appendix Sec.**??**.

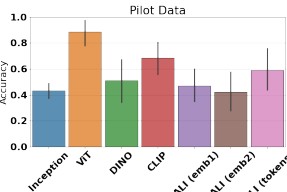 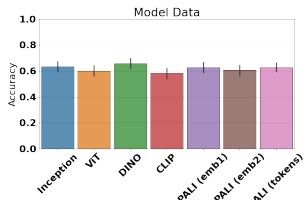 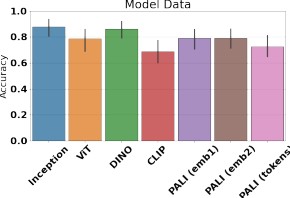

(a) The "diverse" golden set.

(b) Side-by-side model comparisons.

(c) Side-by-side model comparisons with diversity gap > 4.

Figure 6: **Autoevaluation results:** the performance of the Vendi Score given different embeddings across three settings: (a) the golden set; (b) all the annotations gathered; (c) the "easy" subset of the annotations where raters identified a diversity gap of > 4 for a pair. On the golden set, VIT performs best but this does not transfer to side-by-side comparisons. The performance is generally better on the "easy" split of the data, showing that the embeddings perform considerably worse when the difference between the generated sets of images is more subtle—models are more similar.

250 each image in a set. We then use the Vendi Score in order to aggregate embeddings and obtain a
251 diversity prediction for the set. Finally, we consider the first word output by the PALI model as a
252 discrete token. We aggregate these outputs by counting the number of unique words generated for a
253 set to get an estimate for diversity.

254 For each pair of image sets, we analyze the agreement between a diversity assessment based on our
255 autoraters, and the assessment resulting from the human annotations, not taking into account pairs
256 where the annotators found the sets to be equally diverse. If the autoraters and the human evaluations
257 both indicate the same set as being the most diverse (i.e., $s_\Xi(X_1^{j,k}) > s_\Xi(X_2^{j,k})$ and annotators rated
258 the set $X_1^{j,k}$ generated with model 1 based on concept $c^j$ and attribute $a^{j,k}$ as more diverse than $X_2^{j,k}$
259 generated with model 2 based on the same concept and attribute), we say that for that pair of sets,
260 the autorater is correct, else it is incorrect. We then report accuracy by aggregating the number of
261 pairs for which the autoraters are correct. Results are reported in Figs. 6a-6c. We can see that, on
262 the "diverse" golden set, the VIT model does the best, and then the tokens of PALI. This is perhaps
263 surprising, as the VIT model is not specifically trained to focus on the aspects we are considering for
264 diversity but to be able to discriminate between broad classes. However, we see minimal difference
265 in results if we consider the model data. All approaches perform similarly and lead to accuracies
266 that are not significantly different. We hypothesize that the reason for the observed small difference
267 in results was that the human annotators were overall not confident in their ratings. As a result, we
268 looked at ratings where the annotators were confident by using the counts as a proxy. We consider a
269 subset of the data where the difference in counts between the two sets is greater than 4, keeping about
270 24% of the data. We find that now, on the model data we see a bigger difference in results. First, all
271 autoraters are more accurate. Second, we can see that again the image based approaches (e.g., the
272 INCEPTION model, the DINO model and VIT model) perform best.

273 In Figs. 7 and 15 we visualize examples for four side-by-side comparisons where the corresponding
274 autoraters indicate that a group of images have highest or lowest diversity. We can see that results
275 are reasonable and that in general, images with low diversity arise due to mode collapse, i.e. the
276 model generates a very similar image for the same concept. This could explain why the INCEPTION
277 model performs poorly on the pilot data but well on the model comparison data. INCEPTION features
278 are effective for identifying these issues but no effective for identifying diversity in the case of
279 confounding aspects (e.g., the background is changing while the animal is staying the same).

### 3.3 Ranking models with autoevaluation approaches

281 Ranking is achieved by counting the frequency at which the model on the left achieves a higher
282 score than the model on the top (i.e. for "model 1" on the left axis and "model 2" on the top,
283 we count how many times $s_\Xi(X_1^{j,k}) > s_\Xi(X_2^{j,k})$, with $X_1^{j,k}$ generated with model 1, and $X_2^{j,k}$
284 generated with model 2), and subtracting 0.5. The win rate matrices with all models and the score
285 distributions for Imagen 3 and Flux 1.1, the two models that were preferred by human annotators,
286 are shown in Sec. D.5 in the Appendix. In order to test the significance of these comparisons, we
287 aggregate the scores per concept and perform a Wilcoxon signed-rank test under a 95% confidence

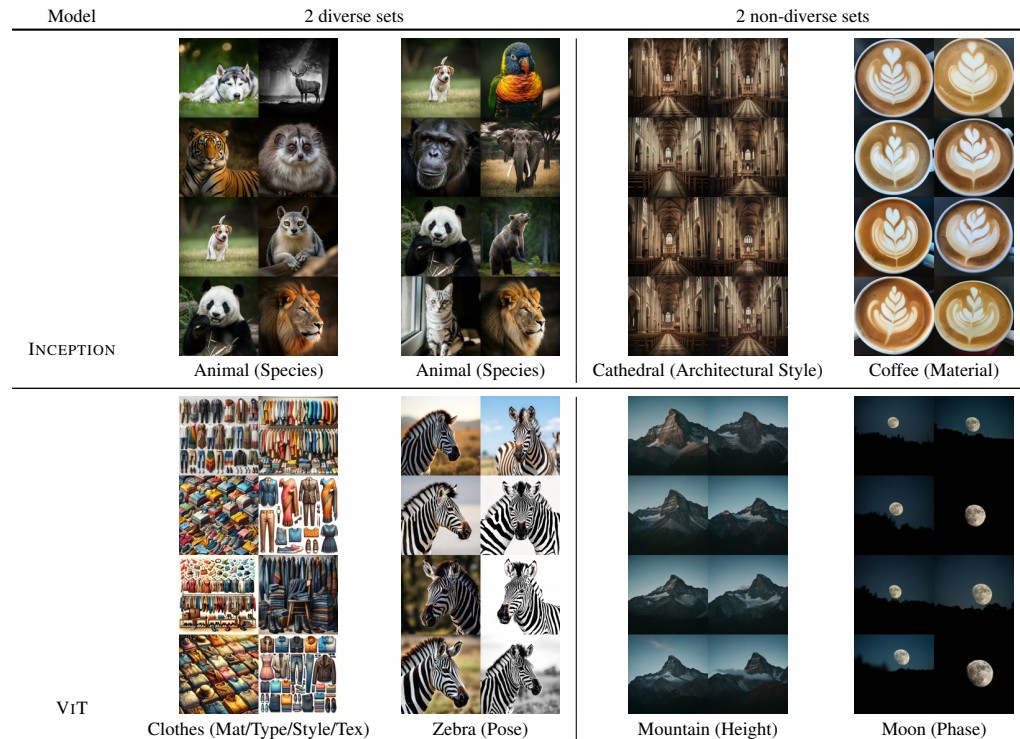

| Model | 2 diverse sets | | 2 non-diverse sets | |
|---|---|---|---|---|
| INCEPTION | Animal (Species) | Animal (Species) | Cathedral (Architectural Style) | Coffee (Material) |
| VIT | Clothes (Mat/Type/Style/Tex) | Zebra (Pose) | Mountain (Height) | Moon (Phase) |

Figure 7: Qualitative results for different autoraters on the T2I annotated dataset, showing two very diverse and two non diverse sets as determined by the autorater.

level. On the left panel, we consider the IMAGENET INCEPTION embeddings, as they yielded the highest accuracy on the model data. In the middle and the right panels, we consider text-conditioned embeddings, as they are closest to our human evaluation procedure. We show the results using PALI(EMB1), as they show a marginal advantage on model data. On the middle panel, we show the results corresponding to conditioning the embedding model on the attribute only, while on the right panel, conditioning takes into account both attribute and object. Results with other embeddings can be found in the Appendix (Sec D.5). Through the autoevaluation model ranking, we find that independently of the chosen embedding, Imagen 3 is not worse than all other models, and Flux 1.1, Imagen 3 and DALLE3 are better than Imagen 2.5 and Muse 2.2. We also observe that using the IMAGENET INCEPTION embeddings and the PALI(EMB1) with a conditioning on object and attribute captures more differences across the three top models, and that using both types of the PALI(EMB1) embeddings captures more differences between Imagen 2.5 and Muse 2.2.

By adopting the model comparison results obtained with the human annotations as shown Fig. 5b as ground-truth, we find that all used embeddings are of similar quality in terms of closeness to human perception of diversity. They all did not flip conclusions, but the autoevaluation approach seems more sensitive to certain variations depending on the choice of embedding model and conditioning. Text conditioning, while closest to the human evaluation procedure, did not show a significant advantage with the current choice of embedding models and conditioning. However, we observe in Fig. 8 the influence of the conditioning. The additional results in the Appendix (Sec. D.5) show the influence of the choice of embedding models. It is possible that better choices of models and conditioning prompts can lead to better results, but we leave this question open for future investigation.

## 4 Related work

The primary method for evaluating text-to-image models involves gathering human judgments on a specific benchmark (i.e., a set of prompts). Previous research highlights that the composition of this benchmark significantly influences the resulting model rankings. This has led to the development of benchmarks with broader skill coverage, e.g., text rendering and spatial reasoning [6, 21, 42], as well as benchmarks targeting specific skills like numerical reasoning [17]. Although human evaluation remains the gold standard, numerous automatic metrics have been proposed to potentially

|  | Flux 1.1 | Imagen 3 | DALLE3 | Muse 2.2 | Imagen 2.5 |
|---|---|---|---|---|---|
| Flux 1.1 | × | < | = | > | > |
| Imagen 3 | > | × | > | > | > |
| DALLE3 | = | < | × | > | > |
| Muse 2.2 | < | < | < | × | = |
| Imagen 2.5 | < | < | < | = | × |

(a) Inception embeddings.

|  | Flux 1.1 | Imagen 3 | DALLE3 | Muse 2.2 | Imagen 2.5 |
|---|---|---|---|---|---|
| Flux 1.1 | × | = | = | > | > |
| Imagen 3 | = | × | = | > | > |
| DALLE3 | = | = | × | > | > |
| Muse 2.2 | < | < | < | × | > |
| Imagen 2.5 | < | < | < | < | × |

(b) PALI(emb1) embeddings - conditioned on attribute.

|  | Flux 1.1 | Imagen 3 | DALLE3 | Muse 2.2 | Imagen 2.5 |
|---|---|---|---|---|---|
| Flux 1.1 | × | < | = | > | > |
| Imagen 3 | > | × | > | > | > |
| DALLE3 | = | < | × | > | > |
| Muse 2.2 | < | < | < | × | > |
| Imagen 2.5 | < | < | < | < | × |

(c) PALI(emb1) embeddings - conditioned on object and attribute.

Figure 8: **Ranking by autoevaluation.** We compare model pairs given the Vendi Score based on (a) Inception, (b) PALI(emb1) conditioned on the attribute, and (c) PALI(emb1) conditioned on object and attribute. Each entry in a grid represents a comparison between two models. Significance is tested via the Wilcoxon signed-rank under a 95% confidence level. The sign indicates the model in the row is better ($>$), worse ($<$), or not significantly different ($=$) than the model in the column.

replace human judgments, at least for certain applications [e.g., 13, 42, 15, 23, 34]. Rigorous validation of these metrics is crucial across diverse conditions, including different prompt sets, human evaluation templates, and models [42]. An important facet of evaluating text-to-image models involves measuring the diversity of their output [9, 40]. This has resulted in different metrics, both reference-based [32, 14, 33] and reference-free [10, 30, 25, 27, 22]. The advantage of reference-free metrics is their independence from a ground-truth set, which permits the evaluation of diversity in broader contexts. One such recent metric, the Vendi score [10], has influenced subsequent research [18, 12, 16]. Despite these developments, none of the proposed metrics have undergone thorough evaluation, frequently being tested only on generic prompts or in simplified settings. Moreover, surprisingly, the majority of previous studies lack human evaluation to demonstrate the validity of these metrics. To address this gap, we introduce a prompt set designed for evaluating diversity across particular attributes and propose and validate a human evaluation template to gather ground-truth diversity judgments. Finally, we compare existing metrics and models under various conditions.

# 5   Discussion

Ensuring diversity in text-to-image (T2I) model outputs is essential, serving as a measure of their ability to express real-world variety. However, rigorous evaluation of this diversity, particularly for specific attributes, remains challenging. This paper introduces a novel framework for attribute-specific T2I diversity evaluation. It comprises a systematic prompt set and a human evaluation template, which has been validated to significantly improve the accuracy of human judgments by explicitly defining the attribute of interest. This framework provides a crucial ground truth for understanding and measuring diversity beyond general impressions.

Applying this framework, we ranked prominent T2I models based on their attribute-specific diversity, identifying Imagen 3 and Flux 1.1 as strong performers. Furthermore, we leveraged our human data to evaluate automated evaluation approaches based on the Vendi Score. Our results demonstrate that the choice of embedding space, upon which autoevaluation metrics operate, is crucial for achieving results that broadly align with human judgments. Notably, our findings indicate that Vendi Score-based autoevaluation approaches can capture human-perceived diversity with approximately 80% accuracy and correctly yield similar results for pairwise model comparisons when a comparable statistical analysis methodology is employed. The proposed framework and our collected data are intended to encourage future work on both T2I model improvement and the development of more reliable evaluation metrics. The broad impact of this work lies in its potential to improve T2I model quality in terms of diversity by providing an evaluation framework grounded in human perception. Moreover, unlike the previous work that often relies on attribute classifiers (e.g., gender), our evaluation methodology can be employed to measure demographic diversity in a classification-free manner. This potentially contributes to the development of more responsible AI systems.

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
