# Supplementary Material

## A  Human evaluation task details

### A.1  Instructions

Before completing the annotation task, annotators were given a comprehensive set of instructions including the following guidelines:

- The goal of the task is to compare the how diverse two sets of images are with respect to a given attribute;
- For the given two sets of images, answer the question about how diverse the concept is with respect to the specific attribute highlighted in the prompt;
- You should count how many different instances of a particular attribute they observe on the left and right sets of images, separately;
- For example, if the attribute is "background" and the prompt is "animal", raters should count how many different backgrounds appear in each set of images and finally judge how diversity of the two sets compares to each other with respect to this attribute;
- Finally, based on the counts, pick one of the following options: (1) Left is more diverse; (2) Right is more diverse; (3) Equally diverse; (4) Unable to answer.

Along with the written instructions, annotators were also given examples corresponding to options 1, 2, and 3.

### A.2  Additional information

In total, 24591 annotations were collected in our study, including the pilot runs. The average time to complete the task with the final template was 32 seconds.

## B  Human evaluation template

### B.1  Golden set concept-attribute pairs

The concept - attribute pairs used for the golden set and the validation of the human evaluation template include: <color, flower>, <material, container>, <color, language>, <background, animal>, <material, chair>, <side dish, cookie shape>, <pattern, clothing>, <style, building>, <weather, biome>, <color, vehicle>.

### B.2  User interface screenshots

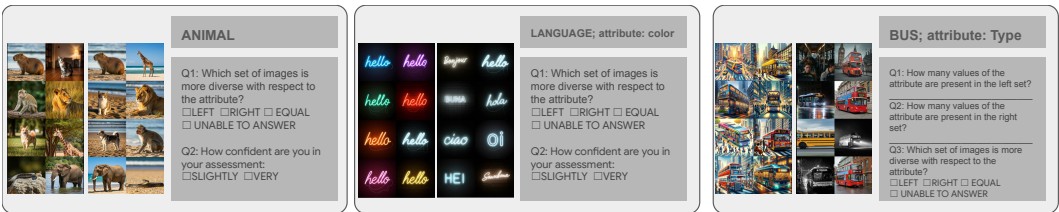

Figure 9: Examples of human evaluation templates used in the pilot study. In the template variant w/o aspect, only the category is provided. In the variant with count, an additional question is included for each set, prompting annotators to specify the number of distinct values observed for the target attribute within the corresponding image set. For exact examples see Figs. 10-12.

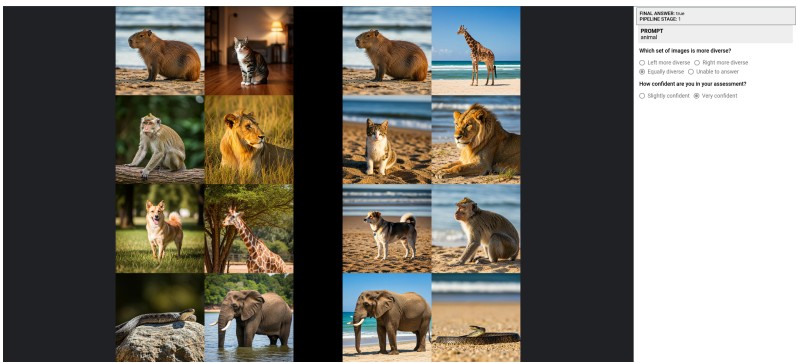

Figure 10: A screenshot of the user interface for one annotation example for the condition "No aspect".

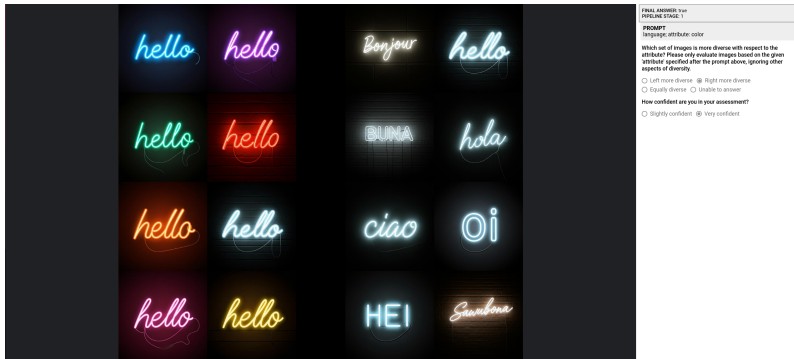

Figure 11: A screenshot of the user interface for one annotation example for the condition "Aspect".

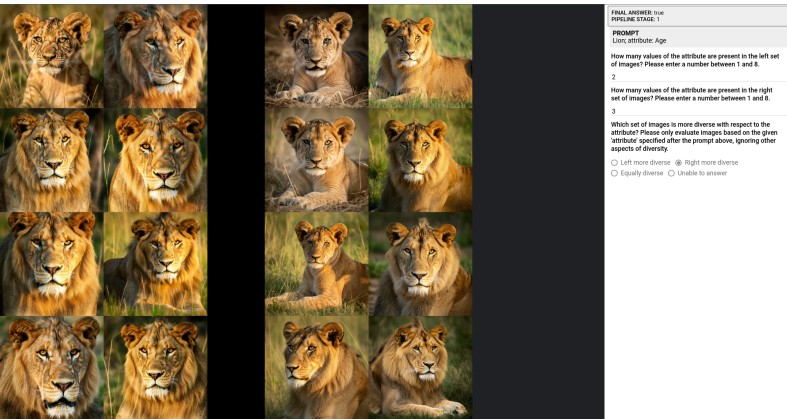

Figure 12: A screenshot of the user interface for one annotation example for the condition "Count".

## C  Additional human evaluation results

 In Fig. 13 we show the histogram of counts averaged across the 5 raters each set in all side-by-side comparisons.

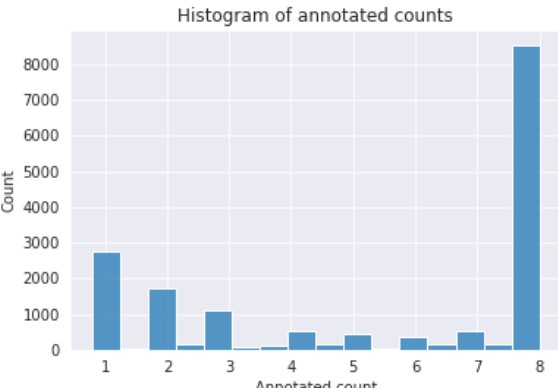

Figure 13: Distribution of all counts annotated by human raters.

## D  Additional autoevaluation results

### D.1  Compute Usage

We used accelerators for running automatic evaluation metrics and generating the images. We run all metrics on a TPU V3 hardware[2]. The image generation pipeline ran on 4 TPUs.

### D.2  Performance for detecting equally diverse image sets

We evaluate how good embeddings are at detecting equally diverse image sets. To not have a threshold-dependent metric, we use the area-under-the-ROC curve (AUC). We construct the true binary label as whether the image sets are labelled as equally diverse or not. We construct the scores as the absolute difference between the metric scores. We then plot the AUC. A good metric would have an AUC close to one, indicating that when the differences are small, the image sets are more likely to have been labelled as the same by the human annotators. We plot results in Figure 14, and find that no metric performs particularly well (AUC < 0.6 in all cases). However, the IMAGENET INCEPTION one performs best, presumably as it is trained to be invariant to small differences and so, as we can see in Figures 7-15, as a lack of diversity usually arises when images are very similar, the embedding performs well. However, we hypothesise that in the face of confounders (e.g. we want to measure diversity of the color of an object but not the type of object), we would not expect such an embedding to do well.

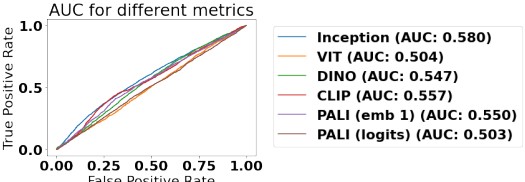

Figure 14: AUC to measure metrics ability to identify sets of equal diversity. It is clear that no metric is particularly effective at differentiating visually similar versus not sets of images.

---

[2]https://cloud.google.com/tpu/docs/v3

## D.3 Additional qualitative results

In Fig. 15 we visualize examples for four side-by-side comparisons where the corresponding autoraters indicate that a group of images have highest or lowest diversity.

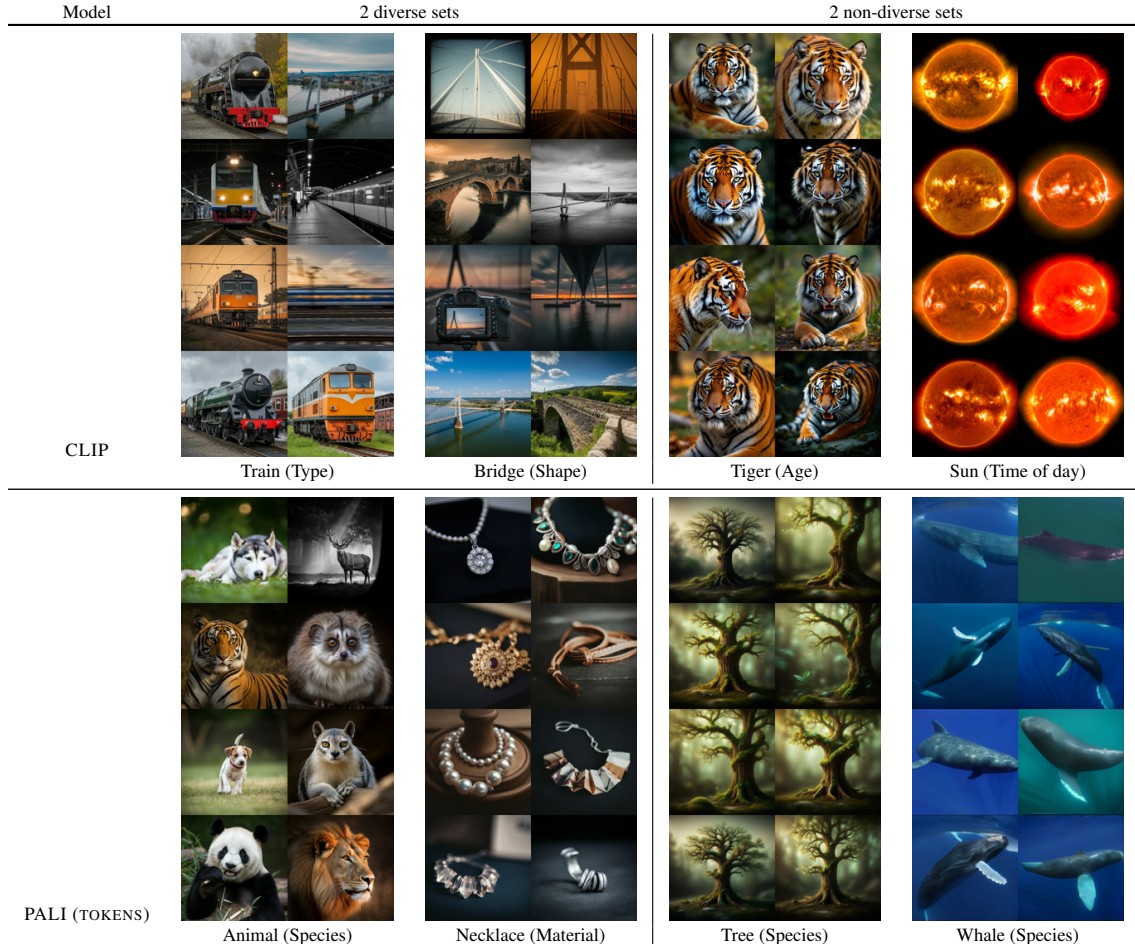

Figure 15: Qualitative results for different models, showing two very diverse and two non diverse sets.

## D.4 Impact of the prompt for the multimodal embeddings

We explore how the choice of prompt impacts results for the multimodal embeddings. We explore four different prompts which differ in their specificity and relatedness to the attributes under question. `[attribute]` and `[object]` are placeholders and filled in based on the object / attribute under test. The templates we consider are as follows:

1. OBJECT_ATTRIBUTE: `What is the [attribute] of the [object]?`
2. ATTRIBUTE: `What is the [attribute]?`
3. OBJECT: `What is the [object]?`
4. EIFFEL: `Where is the Eiffel Tower?`

We would expect the first two questions to be most effective as they directly ask about the property for which we are measuring diversity. The object may be related but can be a confounder and the "Eiffel Tower" question is unrelated.

Results are shown in Figure 16. Surprisingly, we find that we do not see consistent benefit from the two most related prompts (OBJECT_ATTRIBUTE, ATTRIBUTE), implying that the embeddings are

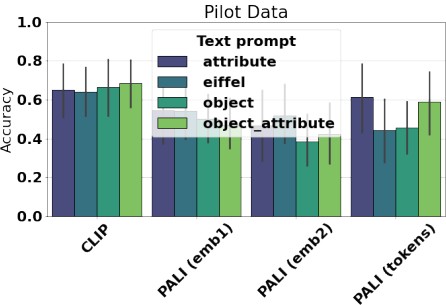
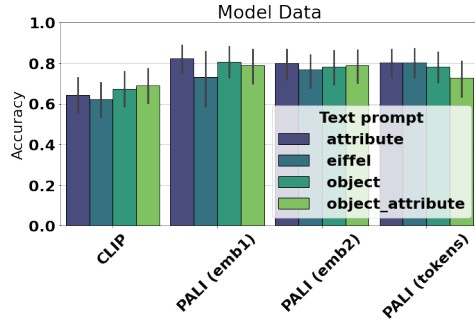

(a) Results on the "diverse" golden set.

(b) Results on the annotation set, where annotators see count differences $> 4$.

Figure 16: Additional auto-eval results that show how results vary based on the textual prompt for the multimodal embeddings. We can see that we *do not* see consistently better results with more related prompts (`What is the [attribute] of the [object]?`, `What is the [attribute]?`), implying the textual input is being ignored.

mostly vision based. A more controllable multimodal embedding we hypothesise would be more effective in this setting.

## D.5 Model ranking with autoevaluation approaches

In this section, we include more results for model ranking based on our auto-evaluation approaches:

- Figures 17, 18 and 19 show the results of compare model rankings in terms of significance in the number of wins with Wilcoxon signed-rank tests under a 95% confidence level using additional models to compute embeddings. This figure completes Figure 8 in Sec. 3.3. In theses figures, we can see:
  - Model ranking based on other embeddings. We observe that similarly to the observations in Sec. 3.3, for all embeddings except IMAGENET VIT, Imagen3 is not worse than all other models. We also observe that independently of the choice of embedding, Flux1.1, Imagen3 and DALLE3 are not worse than Muse2.2 and Imagen2.5. The differences between the models in the top group and the bottom group are more or less detected depending on the embeddings.
  - As mentioned in the main text, we also see the differences between multimodal models. These results highlight how the influence of the choice of embedding models and of conditioning on the model ranking results.
- Figures 20, 21 and 22 show the win rates corresponding to the results shown in Figure 8 in Sec. 3.3 and the additional results described above on the left panels, and compare the distributions of the two best and closest models in terms of behavior according to human evaluation, Imagen3 and Flux1.1, on the right panels. These figures correspond respectively to image models, multimodal model conditioned on attributes, and multimodal models conditioned on objects and attributes.

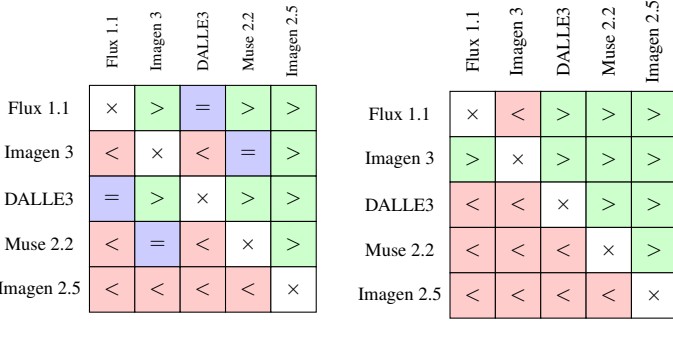

(a) ViT embeddings.

(b) DINO embeddings.

Figure 17: **Model ranking using auto evaluation approaches with additional image models.** We compare model rankings in terms of significance in the number of wins with Wilcoxon signed-rank tests under a 95% confidence level. Each entry in the each of the grids represents a comparison between two models. The $>$ sign indicates the model in the row is better, worse ($<$), or not significantly different ($=$) than the model in the column. The win rates in each of the grids are computed using the scores based on (a) IMAGENET VIT embeddings and (b) DINO embeddings.

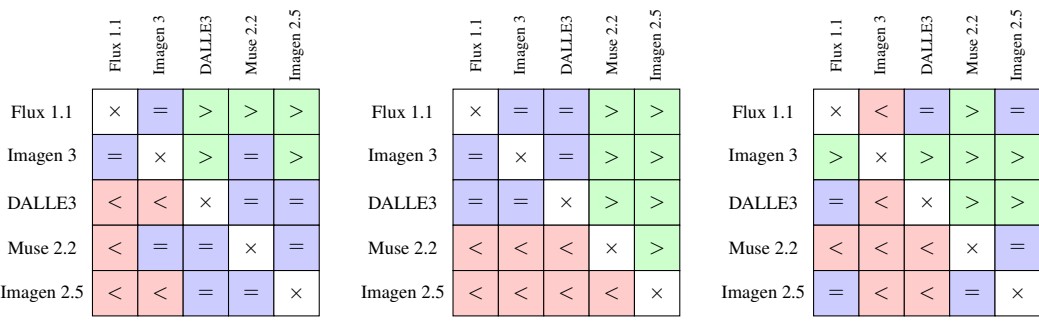

(a) CLIP embeddings.

(b) PALI(emb2) embeddings.

(c) PALI(tokens) embeddings.

Figure 18: **Model ranking using auto evaluation approaches with additional vision and language models conditioned on attributes.** We compare model rankings in terms of significance in the number of wins with Wilcoxon signed-rank tests under a 95% confidence level. Each entry in the each of the grids represents a comparison between two models. The $>$ sign indicates the model in the row is better, worse ($<$), or not significantly different ($=$) than the model in the column. The win rates in each of the grids are computed using the scores based on (a) CLIP embeddings, (b) PALI(emb2) embeddings, and (c) PALI(tokens) embeddings. All models are conditioned on attributes.

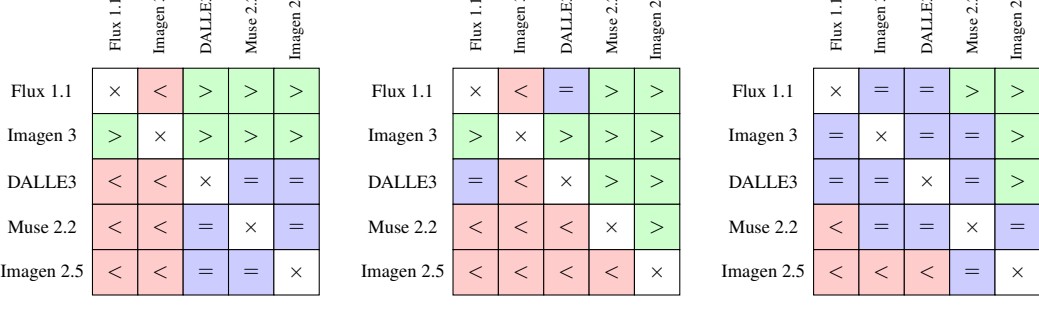

(a) CLIP embeddings.  (b) PALI(emb2) embeddings.  (c) PALI(tokens) embeddings.

Figure 19: **Model ranking using auto evaluation approaches with additional vision and language models conditioned on objects and attributes.** We compare model rankings in terms of significance in the number of wins with Wilcoxon signed-rank tests under a 95% confidence level. Each entry in the each of the grids represents a comparison between two models. The $>$ sign indicates the model in the row is better, worse ($<$), or not significantly different ($=$) than the model in the column. The win rates in each of the grids are computed using the scores based on (a) CLIP embeddings, (b) PALI(emb2) embeddings, and (c) PALI(tokens) embeddings. All models are conditioned on objects and attributes.

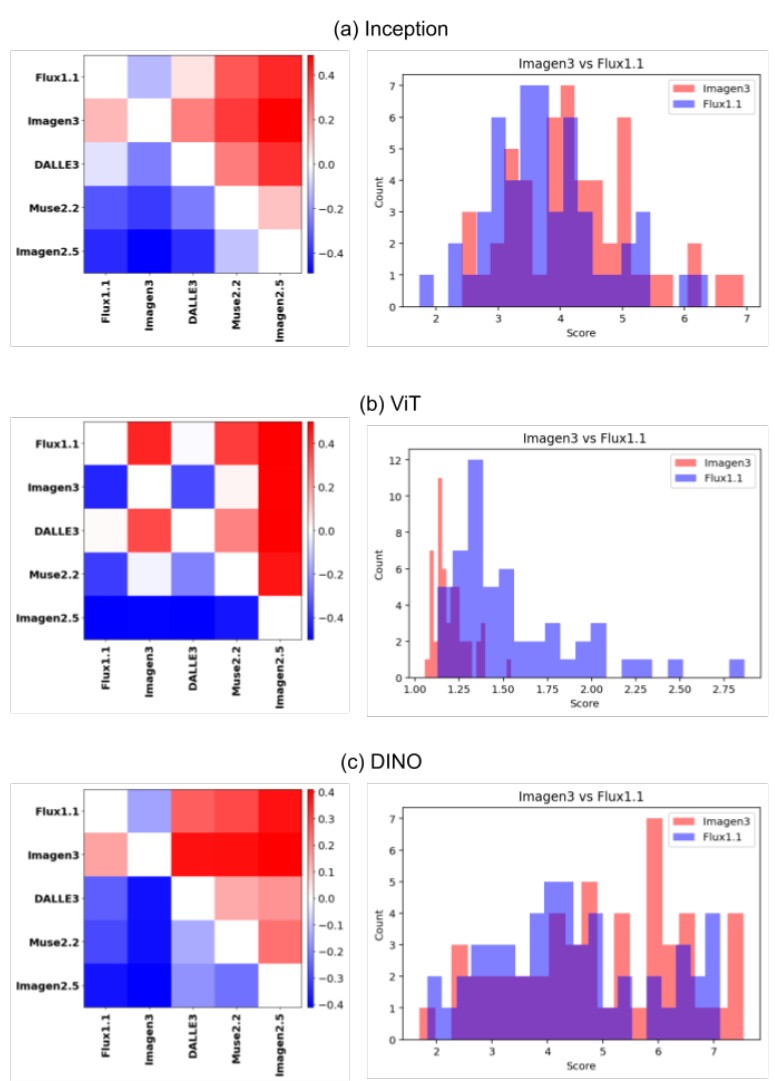

Figure 20: **Model ranking using auto evaluation approaches.** Win rate matrices and score distributions for Flux1.1 and Imagen3 using image models to compute embeddings.

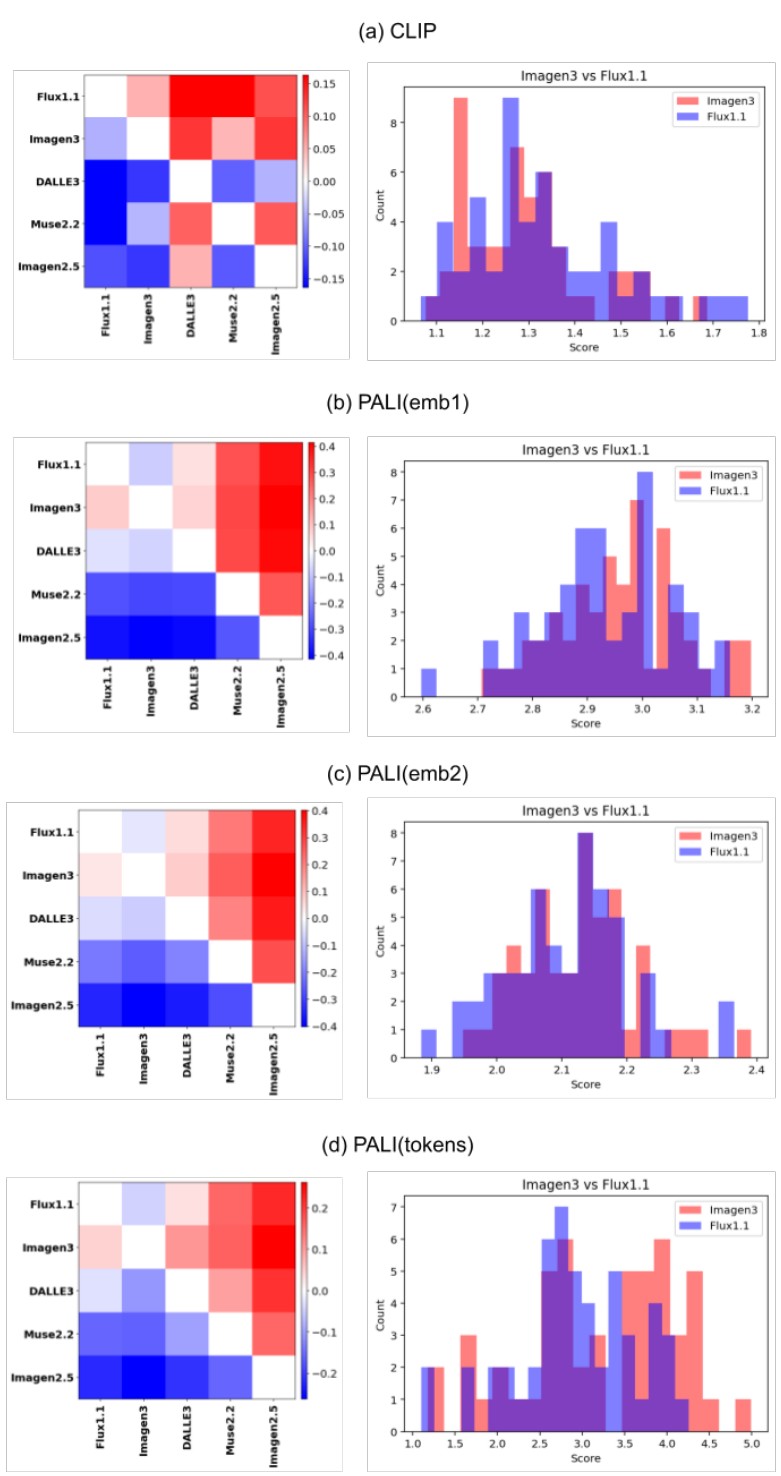

Figure 21: **Model ranking using auto evaluation approaches.** Win rate matrices and score distributions for Flux1.1 and Imagen3 using text-conditioned multimodal models to compute embeddings, conditioned on attributes.

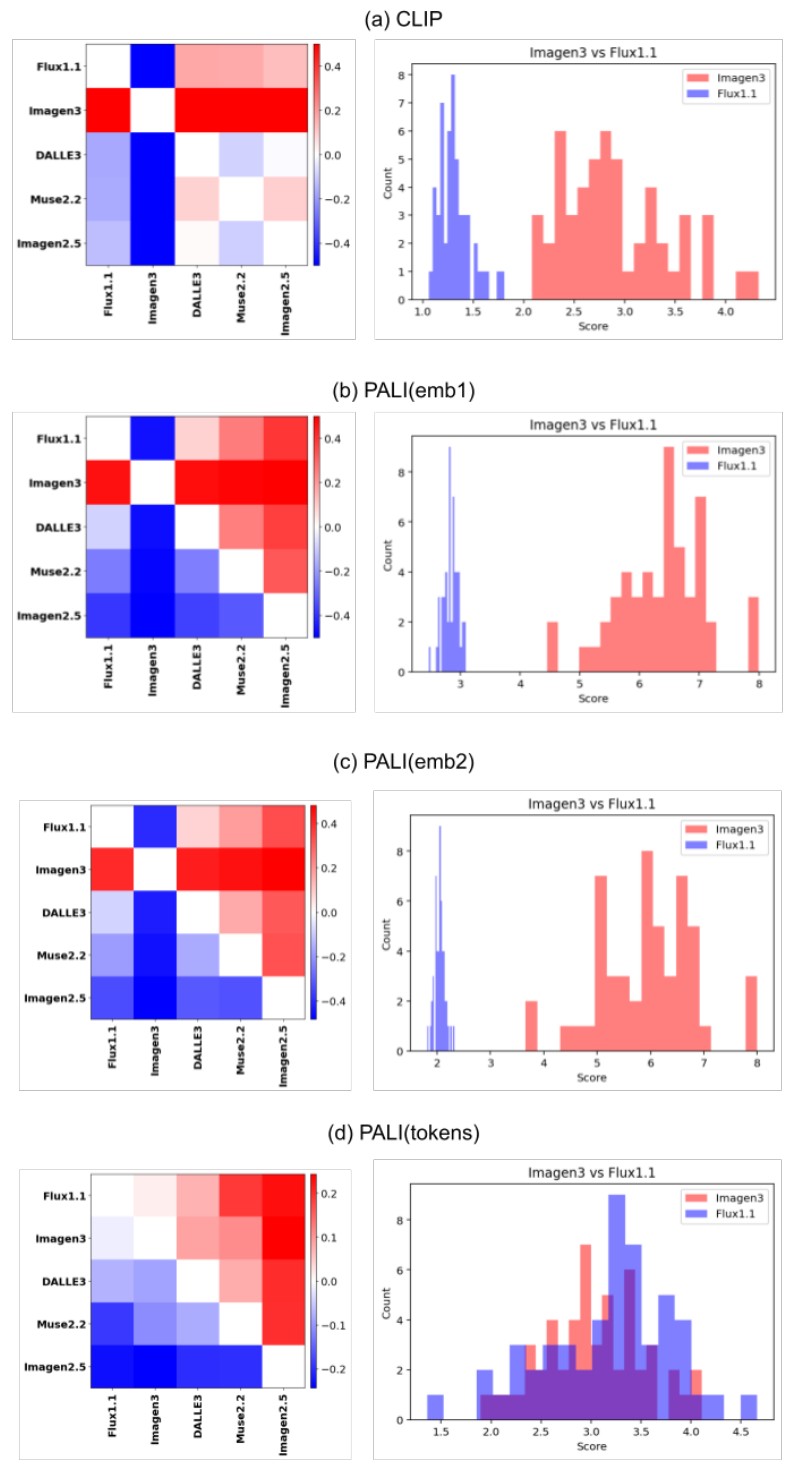

Figure 22: **Model ranking using auto evaluation approaches.** Win rate matrices and score distributions for Flux1.1 and Imagen3 using text-conditioned multimodal models to compute embeddings, conditioned on objects and attributes.

## D.6 Evaluating diversity using foundation models

Besides the investigating multiple embeddings with the Vendi Score for evaluating diversity as presented in Sec. 3, we also propose to use the Gemini model family [37] for comparing T2I models in terms of attribute-based diversity. For that, we use the following instruction: "*I am currently comparing two models with the prompt [prompt] and I would like to know which model generates more diverse images with respect to the attribute [attribute], while disregarding any other attribute in the images. In the following image I show [number of images] images generated by one model in the left, which is [model in the left side] and [number of images] images generated by another model in the right, which is [model in the right side]. You must count the number of different instances of [attribute] in both sets and use this information to decide which set is the most diverse. If there is a set of images which is more diverse than the other with respect to [attribute], can you tell me which one is the most diverse set and explain why? Any other aspects in the images besides [attribute] must not be taken into account. You can also respond that both sets are equally diverse.*" In addition to the instruction, similarly to the human evaluation, two sets of images are given to the model as input.

In Fig. 23 we show the results of three different Gemini models on the task by showing the accuracy in the golden set described in Sec. 2.3. The most recent version of Gemini, v2.5 Flash, achieves the best performance, even surpassing the human raters in this task. These results indicate that such approaches are promising strategies for evaluating diversity which are: (i) able to capture cases where diversity is equally represented in both sets and (ii) do not rely on extracting embeddings.

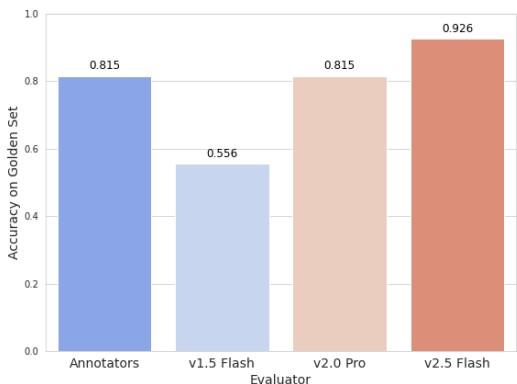

Figure 23: Accuracy of autoraters based on the Gemini model family on the task of comparing diversity of side-by-side sets of 8 images from the golden set. Most recent versions of Gemini perform better in the task, with the v2.5 Flash model surpassing the accuracy of human evaluators.