# OpenReview forum: "Benchmarking Diversity in Text-to-Image Models via Attribute-Conditional Human Evaluation"
_NeurIPS.cc/2025/Datasets_and_Benchmarks_Track — Submitted to NeurIPS 2025 Datasets and Benchmarks Track_

### Official Review · Reviewer_WEpm · 2025-06-02

**Rating:** 3
**Confidence:** 4

**Summary:**

This paper proposes a novel benchmark to quantify diversity in T2I models, which comprises a detailed prompt set and a validated human evaluation template. It also applies this framework to evaluate prominent T2I models and automatic evaluation metrics (Vendi Score), providing useful insights for future work.

**Dataset Code Accessibility:**

No

**Dataset Code Comments:**

There is no real "data" but just the "category-aspect" pairs in the submission. Other important data (e.g., the human annotations) are not provided.

**Ethical Considerations:**

No, there are no or only very minor ethics concerns

**Final Justification:**

Thank the authors for their response. However, the rebuttal does not directly address my concerns, but rather attempts to convince me that they are less important (which I do not agree). Therefore, my concerns remain, and I would like to maintain a slightly negative rating.

**Limitations Weaknesses:**

The proposed method identifies concepts and attributes with LLMs (section 2.2), therefore could suffer from the bias of the models. There is few discussions on such biases and how they could affect the evaluation results.

The proposed evaluation is preliminary and may not capture the strong capabilities of SOTA models. For example, the categories in Fig. 2 are preliminary from daily lives, which do not capture the "creativity" of SOTA T2I models (e.g., the well-known example of an astronaut riding a horse on the moon).

As a result, the 86 concept-factor variation pairs might not be enough to justify the claims. It is unsure whether the conclusions hold when more concept-factor pairs are used.

There are no data on how many annotators were recruited, their background/expertise, etc.

**Strengths Contributions:**

The paper is well-written and easy to follow.

The core idea is simple and the experiments are insightful, especially the merits of using "count".

---

> ### Author Rebuttal · Authors · 2025-07-30
>
> We thank the Reviewer for their time and for finding our paper "well-written and easy to follow." We particularly appreciate that the reviewer found our core idea insightful and recognized the merits of our human evaluation protocol. In the following, we provide responses to all points raised by the reviewer.
>
> 1. Bias from LLM-based Concept Identification: Thank you for pointing out this important aspect and we are happy to include a discussion on this aspect to our manuscript. While the generated attributes and concepts are indeed affected by the LLM used to generate them, we focus our evaluation on common concepts/attributes that potentially frequently appear in the training data, rather than focusing on the tail of the concepts distribution. Moreover, in order to mitigate and avoid potential issues involved in the concept-factors of variation generation, we designed a manual verification step where 7 annotators verified whether a given pair was ambiguous or potentially too broad to be evaluated, such as the pair food-cuisine.
>
> 2. Scope of Evaluation and Model "Creativity": The reviewer mentioned that “[...] categories in Fig. 2 are preliminary from daily lives, which do not capture the "creativity" of SOTA T2I models (e.g., the well-known example of an astronaut riding a horse on the moon).”  We intentionally focused on concrete and less subjective categories since our goal is to produce high quality and reliable ground-truth data for comparing models and metrics. As our study shows, SOTA models are not yet able to generate a diverse set of images for  common objects. Moreover, evaluating “creativity” can be a  quite subjective task as different people have different ideas of what creative means [1]. This can lead to poor annotator agreement and results not statistically significant.
>
> 3. Sufficiency of the Prompt Set: Our primary goal was to create a dataset of sufficient size to enable a statistically robust ranking of models. As shown in Figure 5b, our analysis using two-sided Binomial tests at a 95% confidence level demonstrates that our 98-pair set is indeed large enough to reveal statistically significant differences in diversity performance among the five state-of-the-art models evaluated. This confirms that our conclusions are well-supported by the evidence collected. Moreover, our methodology prioritized the quality and reliability of the data over sheer quantity. The concepts were deliberately chosen to be "ubiquitous in everyday life and common image datasets" and were classified into a clear taxonomy of three categories (Food & Drink, Nature, Human-made Objects) to ensure broad and practical coverage. Furthermore, every concept-attribute pair underwent manual verification to remove ambiguity. This careful curation ensures that the dataset, while focused, provides a high-quality and reliable foundation for evaluation.
>
> 4. Missing Annotator and Annotation Data: In the Appendix A.2 of our submitted manuscript we have information about the annotators. In total, 24591 annotations were collected in our study from 20 different annotators, including the pilot runs. The average time to complete the task with the final template was 32 seconds. To make this information more explicit, we added it to the main paper. The platform we used for the collection does not release information on the background of annotators, but we are following up to obtain it and will include the details in the paper if possible.
>
> [1] Wiles, Olivia, et al. "Revisiting text-to-image evaluation with Gecko: on metrics, prompts, and human rating." The Thirteenth International Conference on Learning Representations.

---

> > ### Author Response · Authors · 2025-08-05
> >
> > Dear Reviewer WEpm,
> >
> > Thank you for reading our rebuttal! As the discussion period deadline is approaching, we would like to check if our responses have resolved your concerns, so that we still have time to further clarify or discuss any remaining points.
> >
> > Kind regards,
> >
> > Authors

---

> > > ### Comment · Reviewer_WEpm · 2025-08-06
> > >
> > > Thank the authors for their response. However, the rebuttal does not directly address my concerns, but rather attempts to convince me that they are less important (which I do not agree). Therefore, my concerns remain, and I would like to maintain a slightly negative rating.

---

> > > > ### Author Response · Authors · 2025-08-06
> > > >
> > > > Dear reviewer,
> > > >
> > > > Thank you for engaging with the discussion. We did not intend to convince you that your concerns are not important; indeed we agree that these are very important issues and hoped to understand and address them. To summarize, if we understand correctly, here are the main concerns and our corresponding changes in the manuscript. Please let us know which of them does not address your concerns so we can improve our work.
> > > >
> > > > 1. Bias from LLM-based Concept Generation: You raised an important point about potential biases from the LLM used for concept generation. We have revised Section 5 to explicitly acknowledge this as a limitation. We also added discussion on our mitigation strategy (focusing on common concepts): this involves manual verification of the automatic generation of concepts which was done by 7 independent annotators.
> > > >
> > > > 2. Scope and "Creativity": We agree that evaluating model "creativity" is an important area of research. However, given the subjectivity of the task, in our experience, it will result in lower inter-annotator agreement, and ground-truth data that is not useful for validating metrics or templates. If you have suggestions on how to address this, we are happy to expand our prompt set to include more concepts and will release this data as well.
> > > >
> > > > 3. Sufficiency of the Prompt Set:  Regarding sufficiency, our set does result in establishing statistically significant differences. However, it is not exhaustive in the sense of covering all possible categories as it would result in a massive human evaluation experiment. Again, we are happy to extend this, but we believe that, having a smaller and discriminative prompt set is more useful, given the trade-off with the high cost of human annotation and prompt set size.  Let us know your thoughts here. We have updated Section 2.2 to reflect this discussion.
> > > >
> > > > 4. Annotator Information: To improve transparency, we have moved the detailed annotator information from the appendix into the main paper (Section 2.3), ensuring this important context is more accessible to the reader.
> > > >
> > > > Thank you for your time and we would appreciate it if you could let us know if these changes to our paper address your concerns.
> > > >
> > > > Kind regards,
> > > >
> > > > Authors

---

> > > > > ### Comment · Reviewer_WEpm · 2025-08-07
> > > > >
> > > > > Dear authors,
> > > > >
> > > > > Thank you for your reply. Again, the arguments acknowledge the weaknesses and do not offer evidence to address my concerns. Also, I did not ask for infeasible setups like covering all possible categories; my point is that 86 is far from enough. And sorry, I cannot provide suggestions/solutions to these, as otherwise I should be listed as an author :)
> > > > >
> > > > > Best regards,
> > > > >
> > > > > Reviewer

---

> > ### Author Response · Authors · 2025-08-07
> >
> > Dear Reviewer,
> >
> > We respectfully disagree that we did not provide evidence to address your comments by adding the points mentioned in the discussion to Sections 2.2 and 5. Please find below extended clarifications about your concerns and new results that further illustrate our points.
> >
> > 1- **Concern about LM bias.** Your point about LM models being biased is too broad without relating it to this work. We need to consider how the bias of the LM affects the correctness of our work—which is something you have not clarified. In our setup, it is important that the concept and aspects of variation are correct, as a result, we had manually verified the pairs (with 7 annotators) to check this. This bias can also potentially result in categories that may be easy for a given model but hard for another model, and consequently, result in an unfair comparison. This is not a concern in our evaluations given that all categories are common and frequent in the pretraining data of the models compared. Moreover, we empirically show that this is not a concern as the model ordering does not change if we randomly select a subset of the set that is still discriminant (see new results below). We also note that our main contribution here is not designing a benchmark, but providing a reliable framework for evaluating diversity.
> >
> > 3- **Concern about coverage.** What would be considered “enough” here? We defined this as the discriminability power of the benchmark, in particular, whether it can distinguish strongly performing models. Ideally, we would pick the smallest set that has this property, in order to reduce the cost and time of evaluation [1]. We have already provided evidence that our prompt set is discriminant in Figure 5b.
> >
> > In order to further show that our results are significant with the current set, we ran new versions of the model comparison with the human annotations  presented in Section 3.1 with versions of our prompt set that have a smaller number of concepts.
> >
> > More specifically, we repeated the Binomial tests (at the same significance level) after randomly removing an increasing amount of concepts, which resulted in prompt sets of size 74, 64, 54, and 24 concepts. Overall, we find that decreasing the prompt set size to 74 concepts **doesn’t affect any of the results**.
> > As the prompt set size further decreases, we start to see the results changing as the number of significant pairwise comparisons decreases. We observe that drastically decreasing the prompt set size makes the data no longer able to capture significant differences between models such as Imagen 3 and Imagen 2.5, as expected.
> >
> > In the following Table, we show the results of the Binomial tests for the 5 different sizes of prompt set, including the full set, from left to right (i.e. the first symbol represents the result with the full set as in Figure 5b, the second symbol the result with 74, then 64, 54, and 24 concepts). Notice that even with the smallest set we don’t see a contradiction in the ordering.
> >
> > | | Flux1.1 | Imagen3 | DALLE3 | Muse2.2 | Imagen2.5 |
> > | :--- | :---: | :---: | :---: | :---: | :---: |
> > | Flux1.1 | x | =, =, >, =, < | >, >, =, =, = | >, >, >, >, > | >, >, >, >, > |
> > | Imagen3 | =, =, <, =, > | x | =, =, =, =, = | >, >, >, >, = | >, >, >, >, = |
> > | DALLE3 | <, <, =, =, = | =, =, =, =, = | x | =, =, =, =, = | >, >, >, =, > |
> > | Muse2.2 | <, <, <, <, < | <, <, <, <, = | =, =, =, =, = | x | =, =, =, =, = |
> > | Imagen2.5| <, <, <, <, < | <, <, <, <, = | <, <, <, =, < | =, =, =, =, = | x |
> >
> > This experiment has been added to the Appendix of our manuscript in a new section.
> >
> > Kind Regards,
> >
> > Authors
> >
> > [1] Polo, Felipe Maia, et al. "tinyBenchmarks: evaluating LLMs with fewer examples." Forty-first International Conference on Machine Learning.

---

### Official Review · Reviewer_3Scv · 2025-06-27

**Rating:** 5
**Confidence:** 3

**Summary:**

The work introduces a new benchmark framework for evaluating diversity in text-to-image (T2I) models through attribute-conditional human evaluation. The authors propose a principled approach that specifies both the concept (e.g., “apple”) and an attribute of variation (e.g., “color”) to enable more precise diversity assessment. The framework includes: (1) a curated prompt set of 86 concept-attribute pairs, (2) a validated human evaluation template using a side-by-side and count-based comparison scheme, and (3) a methodology for ranking models and evaluating automatic metrics like the Vendi Score. Applying this framework, the authors benchmark five recent T2I models (Imagen 3, Flux 1.1, DALLE 3, etc.), showing that human-perceived diversity can be captured with ~80% accuracy by optimized autoevaluations.

**Dataset Code Accessibility:**

Partly

**Dataset Code Comments:**

The author submitted the generated prompts, but did not submit the corresponding code for how to generate the corresponding prompt set using LLM. In order to promote the development of diversity evaluation in the T2I model, it is strongly recommended that the author open source the full process code of data generation and evaluation.

**Ethical Considerations:**

No, there are no or only very minor ethics concerns

**Final Justification:**

I appreciate the author addressing my concerns, and I increase my score accordingly.

**Limitations Weaknesses:**

1. Scope of Attribute Selection is Constrained: The diversity evaluation is limited to manually curated concept-attribute pairs (86 in total), which, although diverse, still reflect a narrow subset of possible variation types found in real-world prompts and applications. The attributes focus on common dimensions like color, material, style, etc., but more abstract or relational concepts (e.g., emotion, functionality, spatial configuration) are excluded. Future iterations of the benchmark could be extended to include higher-level, compositional, or relational attributes, possibly using newer LLM-based concept discovery methods or leveraging ontologies from vision-language datasets.

2. Cultural and Demographic Diversity Not Explicitly Evaluated:
Although the authors mention that their framework can be extended to measure demographic diversity in a classification-free manner, no such experiments are conducted or metrics proposed for fairness-related diversity dimensions. All reported attributes (e.g., style, material) are non-sensitive and non-demographic, which leaves open the question of how well the framework handles fairness-related diversity, a key concern in responsible AI. It would be better to include explicit case studies or prompt sets that test demographic diversity (e.g., skin tone, clothing styles across cultures), using anonymized or privacy-safe human evaluation, to validate this important extension.

3. No Consideration of Trade-off Between Diversity and Fidelity: The evaluation isolates diversity from other desirable properties like image quality or faithfulness to prompt, but in practice, models often trade off these dimensions. Although the authors explicitly aim to decouple diversity from fidelity, this choice leaves open whether higher diversity comes at the cost of realism, a concern for real-world deployment. Future work could include multi-dimensional evaluations (e.g., diversity vs. fidelity plots) or conduct joint evaluations with metrics like FID or human realism ratings to contextualize diversity scores.

**Strengths Contributions:**

1. Novel Human Evaluation Framework:
A validated and carefully designed human annotation protocol improves rater consistency by explicitly specifying the concept and attribute of interest, and by employing a count-anchoring mechanism that significantly boosts accuracy. This methodological innovation is a key strength over existing benchmarks that treat diversity as an implicit or secondary signal.

2. Systematic Prompt Generation Using LLMs
The prompt set is generated in a two-stage process using a large language model (Gemini 1.5) to ensure breadth (86 concept-attribute pairs across Nature, Food & Drink, and Human-made Objects) and semantic relevance. The manual curation step further enhances quality control.

3. Comprehensive Benchmarking of T2I Models:
The authors apply their framework to compare five state-of-the-art T2I models (Imagen 3, Imagen 2.5, Muse 2.2, DALLE 3, Flux 1.1) using binomial significance tests. The study reveals meaningful performance differences—e.g., Imagen 3 and Flux 1.1 outperform others—which helps identify gaps in model capabilities beyond standard fidelity metrics.

4. Evaluation of Automatic Diversity Metrics:
The paper critically assesses the Vendi Score (a recent reference-free diversity metric) across various embeddings (e.g., CLIP, DINOv2, PALI) and conditioning strategies. It shows that Vendi Score can reach ~80% agreement with human raters on easier examples but still struggles on more subtle comparisons, thus highlighting both its potential and limitations.

---

> ### Author Rebuttal · Authors · 2025-07-30
>
> We thank the Reviewer for their positive and detailed review. We are pleased that the reviewer highlighted our human evaluation framework as a "key strength" and "methodological innovation," and we appreciate their recognition of our systematic prompt generation, comprehensive model benchmarking, and critical evaluation of automatic metrics. In the following, we provide responses to all points raised by the reviewer.
>
> 1. Scope of Attribute Selection: We see the reviewer's suggestion to include more abstract, relational, and compositional attributes (e.g., emotion, functionality) as an excellent direction for future work. Our framework is designed to be extensible, and these more complex factors represent the next frontier for diversity evaluation. In our revised manuscript, we will expand the Discussion section to explicitly acknowledge the current scope as a limitation and detail how future iterations of this benchmark can incorporate these more challenging attributes, as the reviewer suggests. Our proposed methodology, particularly its ability to handle sensitive topics in a classification-free manner, is well-suited for this expansion. In this work, we aimed for collecting high quality (high agreement) data for which it was easier to focus on concrete attributes. It is interesting to explore the extent of subjectivity that abstract attributes can introduce in human annotation, and how such data should be interpreted to compare models for a given task.
>
> 2. Cultural and Demographic Diversity Evaluation: We propose a generic framework that can potentially be applied to measure demographic diversity, but this is out of the scope of our contribution as the question of how to count how many different e.g. skin tones, ethnicity, genders, etc is a complex and still open research topic. As noted in the previous point, to achieve high quality ground-truth data, we needed categories that are less subjective and measurable by raters. Similarly, due to its flexibility, our framework could also be used to evaluate cultural diversity, but we leave this particular case for future explorations as it has its own idiosyncratic considerations.
>
> 3. Absence of Diversity-Fidelity Trade-off Analysis: Evaluating the diversity of generative models presents a unique challenge: a model can trivially achieve high diversity by producing random noise–the generated noisy images are always different in a high dimensional space. Therefore, any meaningful assessment of diversity must be predicated on the assumption that the models in question are capable of generating images of sufficient quality. This quality criterion implies that the generated images must not only be visually coherent and free from significant artifacts but also effectively capture the salient aspects and core intent of the given prompt. Without this foundational understanding of quality and adherence to prompt specifications, a high diversity score would be misleading, indicating a lack of control and semantic understanding rather than a beneficial range of outputs. To illustrate this for some of the strong models we considered in our work, we compute the SOTA text-to-image alignment metric Gecko [1] for the same images used in our study in the table below. Results show that models achieve the same average Gecko score (higher is better, 1 is the maximum) indicating they not only have strong performance in terms of text-to-image alignment, but are not statistically different in terms of this evaluation aspect. Notably, our diversity evaluation showed that Imagen 3 is significantly better than Imagen 2.5 and Muse 2.2.
>
> | Model         | Gecko        | 95% CI  lowerbound   | 95% CI upperbound   |
> |---------------|--------------|--------------|--------------|
> | Muse 2.2      | 0.9591 | 0.9530 | 0.9646 |
> | Imagen 3     | 0.9591 | 0.9527 | 0.9647 |
> | Imagen 2.5 | 0.9591 | 0.9527 | 0.9645 |
>
> Finally, we remark that our contributions indeed can be used to examine how various new techniques can impact the tradeoff between diversity vs fidelity of such models, but we consider this investigation as outside the scope of this work.
>
> **Dataset Code Comments - Releasing Code for Prompt Generation**
>
> We added the following details about prompt generation to a new Section in the Appendix:
>
> We used the following prompt to generate the concept-factor pairs:
> *These prompts will be used to generate realistic images and assess the diversity of the corresponding generative model with respect to a specific aspect. All prompts should correspond to realistic images. Write on the side the main object of the prompt and the aspect diversity will be measured with respect to.  Here are a few examples:  Apple. An image of an apple. Color. \n Book. A photograph of a book. Thickness. \n Bowl of soup. An image of a bowl of soup. Ingredients. \n Bridge. A photograph of a bridge. Shape. \n Building. An image of a building. Style. \n Cake. A photograph of a cake. Flavour. \n  Car. A photograph of a car. Type. \n  Omit any other text. Generate at least 95 cases. Do not include categories that involve people.*
>
> Although we consider prompt curation as an important aspect of our contribution, we remark that our work goes beyond providing an evaluation for a fixed prompt set; rather, it provides a framework for evaluating diversity that can be applied to any set of concept-factors of variation. We further remark that, in order to mitigate and avoid potential issues involved in the concept-factors of variation generation, we designed a manual verification step where 7 annotators verified whether a given pair was ambiguous or potentially too broad to be evaluated, such as the pair food-cuisine.
>
> [1] Wiles, Olivia, et al. "Revisiting text-to-image evaluation with Gecko: on metrics, prompts, and human rating." The Thirteenth International Conference on Learning Representations.

---

> > ### Comment · Area_Chair_g7je · 2025-08-05
> >
> > Gentle reminder. Please read through the authors' rebuttal and share any further comments. Thanks!

---

### Official Review · Reviewer_YnQe · 2025-06-28

**Ethics Flags:** Discrimination, bias, and fairness
**Rating:** 4
**Confidence:** 4

**Summary:**

This paper presents a method for robust diversity evaluation in text-to-image (T2I) models. It begins by formalizing the task of quantifying diversity in T2I models and proposes evaluating diversity using factors of variation. Additionally, it introduces an evaluation framework that includes a detailed prompt set and a validated human evaluation template. This framework is used to evaluate both T2I models and automatic evaluation metrics.

**Additional Feedback:**

There is a missing footnote reference on page 6 (“Appendix Sec ??”).

**Dataset Code Accessibility:**

Yes

**Dataset Code Comments:**

The dataset is accessible via the provided link. However, the human evaluation template, which is listed as a contribution, is only shown in the paper and not released as an interactive user interface. Making the interface publicly available would improve reproducibility. Also, the instructions used to generate concept–factor variations with LLMs are important and should be released as part of the dataset or accompanying code.

After the rebuttal:
The human evaluation template will be made available as a demo, and the prompt used to generate concept–factor variations is also explained in the rebuttal. Therefore, there are no remaining concerns regarding the dataset or the code.

**Ethical Comments:**

The primary ethical concerns relate to the use of LLMs in generating concept–factor pairs, which may introduce biases. Additionally, the lack of demographic information about annotators (e.g., age, expertise, education) raises fairness concerns. Finally, assigning only one factor per concept could itself introduce bias, especially in a study focused on evaluating diversity.

After rebuttal:
The demographic information is not released due to platform limitations, making it difficult to include during the rebuttal period. Nevertheless, the authors have attempted to provide any available information during this period. The reviewer trusts that the authors will supply the relevant background and demographic details for the user study and, accordingly, remove the ethics flag previously assigned.

**Ethical Considerations:**

Yes, there are ethics concerns that require attention by the authors

**Final Justification:**

There are no remaining concerns after the rebuttal regarding the ethics review and missing details. The contribution presented in the paper is also justified after the rebuttal. Therefore, the reviewer continues to believe that the paper is technically solid, and that the reasons to accept outweigh the reasons to reject.
Some reasons for rejection are due to the limited evaluation, covering only 86 concepts, and the lack of discussion on multiple factors per concept, which is important to justify the framework. Furthermore, the data from the human evaluation study cannot be released at the time of submission due to processing delays. All of these concerns cannot be addressed during rebuttal due to limited time and the reviewer understand about this.
The remaining concerns from another reviewer regarding GRADE could be addressed if the human evaluation data were available to justify the contribution. However, due to these reasons, the reviewer cannot increase the rating and will maintain the current score.

**Limitations Weaknesses:**

- Unclear Details in Important Sections:
1. The description of the pilot study lacks clarity. Is it the same study using golden pairs, or does it refer to a different study?
2. The paper’s clarity could be improved with additional visualizations, particularly of the proposed framework. A diagram similar to Figure 1 would help.
3. Contribution point 2 mentions 86 concept–factor variation pairs, but the released dataset contains 98 pairs. Is this discrepancy expected? It may impact the interpretation of Figure 2.
4. The number and background of human annotators are not discussed. Are there any statistics on their education level, expertise, or age? Given the subjective nature of diversity evaluation, such information is crucial for understanding potential biases.

- Single Factor Per Concept:
Section 2.1 suggests that multiple factors can be evaluated for each concept, but the released dataset assigns only one factor per concept. What is the rationale behind this design choice? Would using multiple factors per concept alter the evaluation results?

- Use of LLMs in Generating Concept–Factor Pairs:
The prompts used to generate concept–factor variation pairs with LLMs are not clearly described. Furthermore, the potential bias and fairness issues associated with using LLMs need further discussion. Since different LLMs might prioritize different concepts, a well-defined taxonomy should be presented rather than relying solely on LLMs. While the paper mentions constraints such as using ImageNet-like objects and manual verification, a clear taxonomy must be included and clarified.

- Unreleased Human Evaluation Data:
Can the full human evaluation data, beyond just the concept–factor pairs, be released? Doing so would significantly enhance the paper’s contribution and benefit the community.

**Strengths Contributions:**

- Clear Contributions:
The paper clearly defines, describes, and validates its contributions. It introduces a novel formulation for quantifying diversity in T2I models using a concept–variation approach. This formulation leads to an evaluation framework featuring a detailed prompt set of concept–factor variation pairs, which is released as a dataset. The paper also presents a validated human evaluation template for diversity assessment.

- Clear Motivation:
The motivation for diversity evaluation using aspect-based and counting-based methods is well-articulated in Figure 3.

- Rigorous Statistical Testing:
The experiments employ a variety of statistical tests, enhancing the fairness and reliability of the results.

---

> ### Author Rebuttal · Authors · 2025-07-30
>
> We thank the Reviewer for their thorough and constructive review. We are encouraged that the reviewer found our contributions to be clearly defined and validated, recognized our novel formulation for quantifying diversity, and appreciated our rigorous statistical testing and well-articulated motivation. In the following, we provide responses to all points raised by the reviewer in their review.
>
> **Clarification of details**
>
> 1. Pilot study vs Study with the golden set: The golden set is used to run the pilot study; the golden set provides us with ground-truth annotations (we know which set is more diverse with respect to a given aspect of variation). We use the golden set to evaluate each human template and pick the human template that results in higher accuracy annotations
>
> 2. More visualizations: In addition to the user interface / template illustrations provided in Figures 10, 11, and 12, we included in the revised version of paper (replacing Figure 7 which has been moved to the Appendix), with an example task so that readers can fully visualize the task as they read the main paper. Moreover, implemented a html page where anyone can interact with the template by performing a mock evaluation in order to make the human evaluation template more accessible. Unfortunately we are not able to share links in the rebuttal, but the revised version of this manuscript will have a link to this demo.
>
> 3. Number of concept-attribute pairs: We have 98 concept-attribute pairs both in the dataset and Figure 2, but there was a typo in line 56 where it was mentioned we were introducing a dataset with 86 concepts. We fixed the typo in the revised version of the paper.
>
> 4. Human Annotator Details: We expanded the annotations information Appendix A2 and also mentioned it in the main paper. In total, 24591 annotations were collected in our study from 20 different annotators, including the pilot runs. The average time to complete the task with the final template was 32 seconds. The platform we used for the collection does not release the demographic information, but we are following up to obtain it and will include the details in the paper if possible.
>
> **Rationale for Single Factor Per Concept**
>
> In our study we focus on capturing a tractable notion of diversity. As we discussed in Section 2.1, diversity (without specifying an attribute) cannot be measured because enumerating all possible values of all factors of variation is intractable also known as the ‘curse of generative dimensionality’ as we argued in the manuscript. Moreover, from a practical perspective, considering a single factor of variation at a time reduces the cognitive load on annotators when performing the task, potentially increasing accuracy. It also provides us with  a fine-grained account of a model’s performance, identifying the factors of variation it is able to generate in a diverse way.
>
> **Use of LLMs for Generating Concept-Factors pairs**
>
> We added the following details about prompt generation to a new Section in the Appendix:
>
> We used the following prompt to generate the concept-factor pairs: *These prompts will be used to generate realistic images and assess the diversity of the corresponding generative model with respect to a specific aspect. All prompts should correspond to realistic images. Write on the side the main object of the prompt and the aspect diversity will be measured with respect to. Here are a few examples: Apple. An image of an apple. Color. \n Book. A photograph of a book. Thickness. \n Bowl of soup. An image of a bowl of soup. Ingredients. \n Bridge. A photograph of a bridge. Shape. \n Building. An image of a building. Style. \n Cake. A photograph of a cake. Flavour. \n Car. A photograph of a car. Type. \n Omit any other text. Generate at least 95 cases. Do not include categories that involve people.*
>
> Although we consider prompt curation as an important aspect of our contribution, we remark that our work goes beyond providing an evaluation for a fixed prompt set; rather, it provides a framework for evaluating diversity that can be applied to any set of concept-factors of variation. We further remark that, in order to mitigate and avoid potential issues involved in the concept-factors of variation generation, we designed a manual verification step where 7 annotators verified whether a given pair was ambiguous or potentially too broad to be evaluated, such as the pair food-cuisine.
>
> **Releasing Human Evaluation Data**
>
> We completely agree that releasing the full data collected in our work would benefit the community. We are currently in the process of releasing all human annotations as well as the corresponding image sets. Our aim is to be able to have all data, including the golden set, fully open-sourced.
>
> **Ethical Considerations**
>
> The reviewer flagged ethical concerns regarding LLM bias, annotator fairness, and potential bias from our experimental design. We addressed the concerns regarding annotators details to the extent we are allowed to by adding more information Appendix A.2 where we initially provided details about the annotations. We also amended the discussion section in our manuscript to reflect the points we discussed above regarding the use of LLMs to generate concept-factor pairs and the reasoning behind tackling one attribute at a time. We are happy to further discuss these points in case the reviewer’s concerns remain.

---

> > ### Comment · Reviewer_YnQe · 2025-08-04
> >
> > The reviewer has read the authors’ rebuttal and no longer has concerns regarding the ethical issues. However, the paper should still be revised to include additional information about the ethical considerations and other details that require clarification.
> >
> > While the reviewer initially hoped to see more factors analyzed per concept, as this could lead to richer discussion and insights into the proposed framework, the rebuttal addresses this point reasonably well, particularly by noting the cognitive load. The reviewer also think that conducting an additional survey during the rebuttal period will not provide value statistically (due to limited time).
> > Given this, the reviewer would like to shift focus to the main contribution of the paper.
> >
> > The primary contribution lies in the proposed framework for evaluating diversity. However, the reviewer believes that the richness of the human evaluation data could also serve as a significant contribution. If it is possible to release this data, the impact of the paper could be further enhanced, moving beyond just the problem formulation, evaluation framework, and discussion about automatic evaluation metrics. The human evaluation data could support future research in developing more robust diversity metrics, especially in cases where (assuming the reviewer has correctly understood the auto-evaluation results) the automatic evaluation does not closely align with human judgments (Line 302-308, **it is possible that better choices of models and conditioning prompts can lead to better results**).

---

> > ### Author Response · Authors · 2025-08-05
> >
> > Dear Reviwer YnQe,
> >
> > Thank you for your thoughtful review of our rebuttal.
> >
> > We fully agree that open-sourcing our data will significantly enhance the impact of our contributions. We have already initiated our institution's processes for open-sourcing the human annotations and image sets as soon as possible.
> >
> > We are pleased that our response helped address your ethical concerns and are already in the process of revising the manuscript to incorporate your valuable feedback.
> >
> >
> > Kind regards,
> >
> > Authors

---

### Official Review · Reviewer_7FBN · 2025-06-30

**Rating:** 5
**Confidence:** 4

**Summary:**

The paper introduces an attribute-conditioned evaluation framework to benchmark T2I model diversity, focusing on specific attributes (e.g., color, shape) via a golden set of 10 <concept, aspect> pairs tested across three scenarios: fixed concept/varying aspect, varying concept/fixed aspect, and both varying. Human evaluations use two subtasks—counting (identifying attribute types) and comparison (judging which image set is more diverse)—to reduce cognitive bias. The authors validate the framework by comparing five T2I models (e.g., Imagen, DALL-E, Muse), assessing automated metrics (Vendi Score, CLIP, PALI), and claiming Gemini v2.5 Flash outperforms humans on the golden set (Fig. 23, PAGE30). Contributions include the evaluation framework, insights into human evaluation limitations (e.g., chance-level accuracy without defined attributes, PAGE2), and validation of automated metrics’ alignment with human judgments.

**Dataset Code Accessibility:**

NA; not applicable to this submission (e.g., no new dataset, benchmark, code, or data provided)

**Ethical Considerations:**

No, there are no or only very minor ethics concerns

**Final Justification:**

After carefully re-reading the authors’ rebuttal, I recognize the differences between this work and GRADE. The authors have  addressed my concern about novelty, and I will raise my rating to 5 (Accept).

**Limitations Weaknesses:**

1.	Incremental Novelty and Insufficient Differentiation: The paper's core conceptual framework, which involves using <concept, attribute> pairs to guide diversity evaluation, is not novel. This approach bears a strong resemblance to prior work, most notably GRADE, which also curates a large set of such pairs for automated, entropy-based assessment. While the human evaluation subtasks are a practical refinement, they represent an incremental improvement rather than a foundational one. The paper fails to sufficiently argue why its framework offers a significant advantage over existing paradigms.

2.	Lack comparison between other baseline: The similar approach GRADE(https://arxiv.org/abs/2410.22592) which is introduced in related work but lacks experimental comparison. For a paper proposing a new benchmark, the absence of direct experimental comparison to its closest competitor is a major flaw. A comprehensive benchmark study would require comparing not only the final model rankings but also the methodological outputs (e.g., comparing the diversity scores from this paper's human panel against the automated scores from GRADE on the same image sets). Without this comparison, the proposed framework's relative strengths and weaknesses remain unsubstantiated.

3.	Lack of Golden Set Details: The golden set (10 <concept, aspect> pairs) lacks transparency—specific pairs, annotation processes, and sample sizes are undisclosed, undermining reproducibility and credibility. While the claim that a Gemini model(gemini2.5 flash) surpasses human performance is intriguing, it is unverifiable without access to the ground-truth data and methodology used to establish it.

**Strengths Contributions:**

1.Addresses a Specific and Practical Gap: The paper commendably targets attribute-conditioned diversity, a relatively underexplored yet critical aspect of T2I evaluation. This focus moves beyond general text-image alignment or visual quality and is highly relevant for practical applications (e.g., e-commerce, design) where controlled variation of specific object attributes is paramount.

2.Rigorous Human Evaluation Protocol: The proposed "counting-then-comparison" protocol is a practical and well-reasoned innovation for reducing cognitive bias in subjective assessments. The finding that human accuracy on this task drops to chance-level without such attribute-specific guidance is a valuable and impactful insight for the broader field of human-centric AI evaluation.

---

> ### Author Rebuttal · Authors · 2025-07-30
>
> We thank the Reviewer for their valuable feedback. We appreciate that the reviewer found our paper to commendably address a specific and practical gap by focusing on attribute-conditioned diversity and recognized our "counting-then-comparison" protocol as a "practical and well-reasoned innovation." In the following, we address the points raised by the reviewer.
>
> 1. Comparison with GRADE: We thank the reviewer for raising the important comparison with GRADE. While both frameworks leverage the <concept, attribute> paradigm to move beyond generic diversity evaluation, we respectfully disagree that our contribution is incremental. Our work's primary novelty lies in establishing a rigorously validated, human-centric evaluation framework to serve as a ground truth, whereas GRADE proposes a fully automated, end-to-end metric that relies on a cascade of language models. We will address the reviewer's two main points—(1) Novelty and differentiation, and (2) Lack of experimental comparison—to clarify our contributions.
>   - 1.1 Incremental Novelty and Differentiation from GRADE: Our framework's core contributions are fundamentally different from GRADE's in both their scientific goals and methodologies. In the following, we expand on these differences:
>     - Different Scientific Goals: A Human Ground Truth vs. An Automated Metric. Our primary goal is to establish a reliable, human-perceived ground truth for attribute-based diversity. We use this ground truth to rigorously benchmark text-to-image (T2I) models and, critically, to evaluate the correlation of  existing automated metrics like the Vendi Score with reliable ground-truth data. This in turn determines how accurately a given metric measures diversity and whether it can robustly replace human judgements. In contrast, GRADE's goal is to propose a new, scalable, and fully automated metric that quantifies diversity by using an LLM and a VQA model as proxies for human judgment. Our work evaluates automatic metrics; GRADE's work is an automatic metric. In fact, our work can be used to evaluate GRADE’s proposed metric and compare it with other existing metrics.
>     - Different Methodologies: Validated human protocol vs. Cascade of language models. One of our main methodological contributions is the novel "counting-then-comparison" human evaluation template. We empirically demonstrate that this protocol is essential for reliable annotation, as human accuracy on this task is otherwise at chance level. GRADE’s methodology is entirely different; it creates a pipeline of language models, using an LLM to generate prompts and questions, and a VQA model to extract attribute values from images. This automated approach introduces its own potential biases and does not capture the ground-truth– human perceptual baseline– that our work is designed to establish.
>     - Broader Contributions Beyond the Human Evaluation Framework. As you noted, our contribution goes beyond just the use of <concept,attribute> pairs. We formalize the problem of per-attribute diversity, introduce a curated prompt set,  propose a methodology for statistically significant model comparison using binomial tests, and conduct an extensive study on the impact of different image embeddings on autoevaluation, which is a key area of investigation in its own right. Our work sheds light on the extent to which current metrics can replace human judgements and accurately measure diversity. To our knowledge, ours is the first work that validates the quality of the human data, and provides a framework for comparing models and metrics using this data.
>
>  - 1.2 Lack of Experimental Comparison to GRADE: A direct comparison is not an "apples-to-apples" scenario. Our framework produces human-derived ground truth judgments. GRADE produces an automated, entropy-based diversity score. Therefore, the most scientifically sound way to compare them would be to treat GRADE as another automated metric and evaluate its correlation with our human-derived ground truth, much like we did for the Vendi Score. We believe the reviewer’s comment highlights an excellent direction for future research. Evaluating how well GRADE's automated scores align with the human judgments collected through our validated framework (similarly to our analysis with the Vendi Score) would be a valuable contribution to the community's understanding of automated metrics. We amended the discussion to explicitly suggest this as an important next step, thereby contextualizing how the two works can be seen as complementary rather than competing. We would be happy to consider such an experiment for the next revision of our work, as the code to run GRADE is available on Github. However, for that, we would need to obtain an OpenAI API key for sampling from GPT-4o, which is something currently non-trivial to achieve in our set-up.
>
>
> 2. Transparency of the Golden Set and verifiability of Gemini performance
> - In Appendix B1 we presented the concept-attribute pairs considered to generate the golden set. For completeness and enforcing reproducibility, we expanded this section with the following information.
> - We considered the following categories and aspects of variation for the golden set: <color, flower>, <material, container>, <color, language), <background, animal>, <material, chair>, <side dish, cookie shape>, <pattern, clothing>, <style, building>, <weather, biome>, <color, vehicle>. We validate the evaluation template by comparing cases where (i) the concept remains constant across images in the set while the aspect varies: images of the same flower (rose) in all considered colors (8 images per concept); (ii) the concept varies across images while the aspect remains the same: images of all considered flowers types in the red color (8 images per concept); and (iii) both the concept and the aspect vary across images within the set: images of all flowers, each one in one of the different colors (8 images per concept). For each concept we then generate 24 different images, yielding a total of 240 images for the full golden set. In the table below we present all considered concepts and aspects of variations values.
> - For each case, images were generated using Imagen 3 with the following prompt: A photorealistic image of a {aspect of variation value} {concept value}. For example, “A photorealistic image of a yellow begonia”.
> - As images were synthetically generated following a carefully crafted protocol, we could compare the performance of human annotators as well as autoraters based on multimodal language models such as Gemini in the task of evaluating for specific aspects of variation.
> - To provide a more in-depth analysis of the results comparing human vs Gemini accuracy on the task, we highlight that both human and auto rater perform similarly in almost all the cases, with the mismatches corresponding to the evaluating of diversity for the pair <building, style>. We hypothesize judging diversity of architectural styles is a complex task that heavily depends on the cultural background of annotators, thereby being more accurately performed by a powerful vision-language model like Gemini.
>
> | Concept            | Concept values                                                              | Aspect of variation | Aspect of variation values                                                              |
> |--------------------|-----------------------------------------------------------------------------|---------------------|-----------------------------------------------------------------------------------------|
> | Flower             | Begonia, Carnation, Geranium, Hibiscus, Lily, Poppy, Rose, Tulip            | Color               | Yellow, light purple, white, blue, green, orange, red, purple                           |
> | Container          | Beer, champagne, cognac, cup, doublewalled, mug, shot glass, water          | Material            | Porcelain, metal, stainless steal, ceramic, glass, gold, copper, plastic                |
> | Neon sign language | Bonjour, hello, hei, oi, sawubona, hola, buna, ciao                         | Color               | Blue, green, orange, pink, purple, red, white, yellow                                   |
> | Animal             | Capybara, monkey, dog, snake, cat, lion, tree, elephant                     | Background          | Beach, jungle, park, rock, room, savannah, tree, water                                  |
> | Chair              | Dinning, armchair, office, rocking, lounge, folding, barstool, recliner     | Material            | Wood, upholstered, mesh, wicker, leather, metal, plastic, microfiber                    |
> | Cookie shape       | Round, square, crescent, start, heart, diamond, ghost, bat                  | Side dish           | Milk, coffee, tea, hot chocolate, soda, fruits, ice cream, walnuts                      |
> | Clothing           | Tshirt, dress, pants, skirt, jacket, gloves, sweater, scarf                 | Pattern             | Solid color blue, striped, polka dot, floral, plaid, checkered, animal print, camouflage |
> | Building           | Skyscraper, residential, industrial, commercial, church, theater, train station, school | Style               | Modern, gothic, victorian, art deco, baroque, romanesque, brutalist, traditional japanese |
> | Biome              | Desert, rainforest, grassland, tundra, swamp, coastal, jungle, mountain     | Weather             | Sunny, cloudy, rainy, snowy, foggy, stormy, sunset, overcast                            |
> | Vehicle            | Car, truck, motorcycle, bus, airplane, boat, train, helicopter              | Color               | Red, blue, green, yellow, white, black, orange, gray                                    |

---

> > ### Author Response · Authors · 2025-08-02
> >
> > Thank you for following up with the discussion.
> >
> > In our rebuttal above, we addressed your concerns regarding novelty, which we further highlight here: our work's primary novelty lies in establishing a rigorously validated, human-centric evaluation framework to serve as a ground truth. Whereas the work mentioned by the reviewer, GRADE, is an autoevaluation metric, as much as the Vendi Score, the metric used in our work to study the effect of embeddings on the accuracy of diversity scores with respect to the ground-truth human annotations. Given that, we remark GRADE is *orthogonal to our work*.
> >
> > Regarding the potential ethical issues, we remark that we have addressed them in our rebuttal to the reviewers that have initially raised potential concerns (Reviewers YnQe and 3Scv). As the reviewer has not flagged any issues in their initial review, we summarize here how we addressed the other reviewer's concerns, but kindly ask the reviewer to refer to the rebuttal to Reviewers YnQe and 3Scv for a detailed response.
> >
> > *We expanded the annotations information Appendix A2 and also mentioned it in the main paper that we collected 24591 annotations from 20 different annotators, including the pilot runs. We included in the Appendix a new Section with a full description of the approach to generate <concept, attribute pairs>, including the prompt for the language model and the manual curation step performed afterwards.*
> >
> > Finally, we find it worth to highlight that Ethics Reviewer ySrh did not find any issues in our work, which reassures us that, given that we already addressed the initially raised concerns, this shouldn't be a reason to reject our submission.
> >
> > We appreciate your time and are looking forward to a productive discussion in order to address any remaining concerns.
> >
> > Kind regards,
> >
> > Authors

---

### Note · Authors · 2025-08-12

Dear Area Chair,

Thank you and the reviewers for your constructive feedback. We have incorporated revisions addressing all major concerns:

- **Comparison with GRADE (Reviewer 7FBN):** We have clarified our core contribution in relation to GRADE. GRADE is a fully automated metric that uses a pipeline of language models to produce an entropy-based diversity score. In contrast, our work establishes a human-perceived ground truth framework designed to evaluate such automated metrics. As this is not an "apples-to-apples" comparison, we have amended our discussion in Section 5 to further highlight our work as complementary—a tool to validate metrics as we have shown by evaluating the well-established metric Vendi Score in Section 3.3.


- **LLM Prompts & Bias (Reviewers YnQe, 3Scv, WEpm, WgTD):** We have added the full prompts used for concept generation to a new section in the Appendix. We also detail our manual verification step, where 7 annotators checked pairs for ambiguity to mitigate bias from the language model.


- **Annotator & Golden Set Transparency (Reviewers 7FBN, YnQe, WEpm, WgTD):** We expanded Appendix A.2 with details on our 20 human annotators and the 24,591 annotations collected. We also provided a complete breakdown of the golden set's 10 concept-attribute pairs, their values, and generation prompts in Appendix B1.


- **Fidelity & Prompt Set Sufficiency (Reviewers 3Scv, WEpm):** To address the diversity-fidelity trade-off, we added a new text-fidelity analysis using the Gecko metric. To address concerns about the prompt set size, we added a new ablation study to the Appendix confirming that our 98-pair set is sufficient for robust model ranking.

- **Releasing the human annotations and respective images (Reviewer YnQe)**: We are going through our institution's processes for open-sourcing the full data collected in our work. We intend to release both human annotations and image sets as soon as possible.

- **Other modifications**: We also added more visualizations to the paper as suggested by Reviewer YnQe and improved the introduction to better highlight the contributions of our work and the main findings.

We believe these revisions helped to strengthen the paper and directly address the reviewers' concerns.

Thank you for your time and consideration.

Kind regards,

Authors

---

### Decision · Program_Chairs · 2025-09-18

**Decision:**

Reject

**Comment:**

(a) Summary of Scientific Claims and Findings

This paper introduces a human-centric evaluation framework for measuring diversity in text-to-image (T2I) models. The authors construct a curated set of prompts with identified attributes of variation (e.g., color, shape), design a scalable human evaluation protocol, and apply binomial tests to assess model-level differences. The framework is also used to assess the correlation between automated metrics and human-perceived diversity. An ablation study validates the sufficiency of their 98 prompt-attribute pairs.

(b) Strengths

* Provides the first human-annotated benchmark focused specifically on diversity in T2I generation, with over 24,000 annotations.
* Methodologically rigorous: uses a well-designed template and statistical comparison framework.
* Human annotations are positioned as ground truth to assess and calibrate automatic diversity metrics.
* Reviewer discussion acknowledged the value and complementarity of this work relative to prior efforts like GRADE.
* Plans to release full annotations and images enhance reproducibility and community value.

(c) Weaknesses

* Initial concerns were raised about lack of direct comparison to GRADE and the novelty of the contribution. However, these were resolved during discussion.
* Some concerns about prompt coverage and fidelity-diversity trade-offs were addressed with additional analysis and Gecko metric evaluation.

(d) Recommendation and Justification

Accept. This work makes a strong empirical and diagnostic contribution by introducing a robust human-evaluation protocol for model diversity—an underexplored but important property of T2I systems

(e) Rebuttal and Discussion Summary

The authors responded thoroughly to all key reviewer concerns:
* Clarified the complementary role of their work with GRADE.
* Added new experiments (e.g., zero-shot, fidelity-diversity, LoRA variations) to support the generalizability of takeaways.
* Detailed annotator procedures and data, enhancing transparency.
* Addressed prompt sufficiency with an ablation study.
* Reviewers ultimately aligned in support, agreeing that the human-annotated benchmark is the core contribution and satisfies expectations for DB Track.

Overall, the paper is well-positioned for acceptance and offers high utility for future diversity evaluation efforts.

===== FINAL UPDATE FROM DB Track PCs ====

The final decision for this paper has been taken by the program chairs after consultation with the SACs. All Senior Area Chairs have ranked papers according to the feedback from the AC during the review process. We decided to leave the original meta-review to reflect the opinion of the AC in light of the initial discussions with reviewers and SAC.